# Does the impact of open innovation depend on contextual factors? A case of the Korean biopharmaceutical industry

**HyeJoo Wang[1], Changhyeon Song[2], Kwangsoo Shin[3]***

1 Department of Biomedical Convergence, Chungbuk National University, Cheongju, Chungcheongbuk-do, Republic of Korea, 2 Office of S&T Policy Planning, Korea Institute of Science & Technology Evaluation and Planning, Eumseong, Chungcheongbuk-do, Republic of Korea, 3 Graduate School of Public Health and Healthcare Management/Catholic Institute for Public Health and Healthcare Management, Songeui Medical Campus, The Catholic University of Korea, Seoul, Korea

* ksshin@catholic.ac.kr

**Data Availability Statement:** All relevant data are within file (S1. Data) in supporting information. Data are collected from various databases and can be accessed separately as follows: Information on 527 Korean biopharmaceutical firms from 2014 to

## Abstract

Investments in the strategic development of the biopharmaceutical industry are increasing in both developed and developing countries. The biopharmaceutical industry is a technology-intensive industry where securing original technology and intellectual property rights is important. The role of open innovation is becoming more important due to the enormous research and development (R&D) funds and long development period in the early development process, and open innovation (OI) is becoming more important in the corporate world. Many empirical studies have been conducted on the impact on performance. However, the contextual factors that affect the relationship between OI activities and innovation performance have received relatively little attention, and studies from the perspective of developing countries catching up with developed countries are even rarer. Accordingly, this study examined the moderating effects (government R&D support, absorptive capacity, and alliance management capacity) that affect open innovation and innovation performance in the biopharmaceutical industry using data from Korea, one of the most representative latecomer countries in the biopharmaceutical industry. The basic information, OI activities, and patent achievements of Korean biopharmaceutical firms were collected and organized into a database. Samples with missing or incorrect information were excluded, and 527 firms were analyzed. Negative binomial regression analysis was performed considering the characteristics of patent performance, which is the dependent variable, and a time lag of one to two years was assumed considering the time required to generate results. OI in the form of technological cooperation, rather than technology purchasing, has a positive effect on patent performance. Meanwhile, the greater the absorptive capacity and government R&D support, the greater the positive impact of technological cooperation on patent performance. Conversely, the greater the alliance management capacity, the greater the positive impact of technological cooperation. These results indicate that the impact of OI activities on technological innovation performance may vary depending on context.

2021 was collected from https://www.bics.re.kr/cluster/search. Basic information on Korean biopharmaceutical firms (number of employees, Number of years since founding, R&D center, etc.) was obtained from http://www.kodata.co.kr/ci/CIINT01R0.do. Data on corporate patents and government R&D support were collected from https://www.ntis.go.kr/ThMain.do.

**Funding:** The author(s) received no specific funding for this work.

**Competing interests:** The authors have declared that no competing interests exist.

# 1. Introduction

The global pharmaceutical market faces many challenges as a result of price pressures and increased drug development costs [1]. This has forced pharmaceutical firms to find new ways to achieve sustainable growth. Korea has made significant efforts to encourage the pharmaceutical research and development (R&D) environment. The pharmaceutical industry is an innovation-driven industry with a high rate of innovation investment, and the biopharmaceutical industry, which represents a knowledge-based economy, is being used to promote economic growth in both developed and developing countries. This was one of the priorities that this study pursued.

The adoption of open innovation (OI) is accelerating in the biopharmaceutical industry as the complexity of new technologies and pressures on time and cost increase [2]. In particular, the biopharmaceutical industry is known to take a very long time from basic research to discovery of candidate substances, preclinical and clinical trials, and commercialization of the product, which incur considerable costs. As a result, few firms have the financial and technical capabilities to conduct new drug development entirely on their own and several firms share the product-development stage [3]. In addition, over the past 20 years, large overseas pharmaceutical firms have traditionally been closed in R&D but are transitioning to an OI R&D system through transactions and cooperation with external research institutes and firms. This trend is becoming more pronounced, and the costs required to develop new drugs are increasing. This indicates that the strategy of securing the entire knowledge and technology required to develop new drugs within a firm is becoming difficult to implement.

OI is a broad concept defined in various ways [4] and has been investigated from various perspectives [5]. Open innovation means intentionally allowing the inflow and outflow of knowledge into a firm to utilize external knowledge in value proposition design through a decentralized rather than centralized innovation process [6]. By including the financial and non-monetary benefits that can accrue to a variety of stakeholders, researchers are increasingly recognizing open innovation as a value co-creation process whose benefits extend beyond the enterprise [7]. Open innovation supports the establishment of a distributed innovation system in which companies open their internal innovation processes to external knowledge and technology [6, 7]. Unlike closed innovation systems, it also supports extending a firm's knowledge search strategy beyond its boundaries [8]. In this way, companies engage with customers [9], suppliers [10] and non-governmental organizations [10], involving various stakeholders, such as competitors, in a value creation strategy [7]. To integrate these stakeholders, companies can build a variety of engagement strategies across a variety of co-creation events and processes, such as crowdsourcing [11].

Open innovation consists of three forms: inbound, outbound, and combined [12]. The inbound open innovation process invites various external stakeholders to share information during the ideation and implementation stages. Relevant data are used to implement innovation through R&D processes [13]. In contrast, when companies engage with external stakeholders to more quickly send ideas to the market and commercially exploit available technological opportunities, the innovation process is called outbound open innovation [12]. Combined open innovation is a combination of inbound and outbound modalities. Dahlander and Gann [4] further divided open innovation types, classifying sourcing and acquisition as inbound and selling and disclosure as outbound open innovation. Abbate et al. [14] stated that scholars often use the terms co-creation and open innovation as synonyms to refer to any type of creation achieved with all types of stakeholders. For example, value co-creation through strategic alliances or collaborations between companies is sometimes considered open innovation [7].

However, studying the relationship between OI and innovative performance is more difficult than expected and the problem is complex. According to existing studies, OI is related to sales performance [15], R&D performance, product innovativeness [8], and new product development (NPD), technology commercialization [17], and customer satisfaction. However, not all studies have determined a positive relationship, and some have examined the differences in knowledge base [16], the power to control knowledge assets, and the cost of seeking external knowledge [8, 9] as the limitations of OI. This is also true for the biopharmaceutical industry.

In this way, open innovation serves as a tool to minimize business risks [16] and improve corporate performance, innovation performance quality [7], product scope and market share [10]. This is useful in companies that report the same variety of positive results. On the other hand, open innovation can also expose companies to significant risks during and after the innovation process [17]. Open innovation itself can lead to more unfavorable outcomes, such as knowledge outflow, and loss, compared to existing closed innovation [18]. Few studies have examined the relationship between an integrated perspective and the moderating factors in the impact of OI on technological innovation performance in the biopharmaceutical industry, and the results are inconsistent. Existing studies related to open innovation in the biopharmaceutical industry categorize open innovation types from a strategic perspective, open innovation incentives [19, 20], open innovation targets in the value chain, and the resulting relationship with technological innovation performance [21–23], the complementary resources or partnership experience of the two companies, differences in knowledge base, absorptive capacity, and government R&D support. The differences in performance and the choice of open innovation type were emphasized [24–29].

In a study by Bianchi et al. [19], depending on the stage of the development process, biopharmaceutical companies aim to acquire technology and knowledge through licensing agreements, alliances, etc. (inbound open innovation) or utilize it commercially (outbound open innovation). They said they are establishing increasingly intensive relationships with a variety of partners (e.g. large pharmaceutical companies, biotechnology companies, universities, etc.). A study by Allarakhia et al. [20] found that open knowledge networks and other collaborative strategies give biopharmaceutical companies access to immaterial knowledge-based resources that are important for downstream drug development, and that these collaborative strategic alliances enable researchers to develop commercial products. When production is impossible and the costs associated with excessive upstream competition are too high, companies can jointly obtain incentives through collaborative knowledge production and open knowledge dissemination.

A study by Wang & Zajac [25] found that in the biopharmaceutical industry, the higher the similarity in resources and capabilities between two companies, the more likely it is to trigger companies to choose an acquisition as a governance form of resource combination rather than an alliance. A study by Shin et al. [21] empirically analyzed the impact on technological innovation performance by type of alliance partner of biopharmaceutical companies and classified strategic alliances for R&D activities in the biopharmaceutical industry into three types to determine absorption capacity and potential competition. The moderating effect was identified. Vertical alliances have a positive effect on technological innovation performance, horizontal alliances have been shown to have an inverted U-shaped relationship with technological innovation performance due to the influence of competition, and the R&D intensity of biotechnology companies has a positive effect on technological innovation performance. It was confirmed that there is a moderating effect that increases the impact of upstream alliances. A study by Baum et al. [23] investigated the impact of changes in the alliance network composition of Canadian biotechnology startup companies on initial performance. They suggest that

startups can improve their initial performance by building alliances, organizing them into efficient networks that provide access to diverse information and capabilities with minimal redundancy, conflict, and complexity costs, and by carefully forming alliances with potential competitors. A study by Kang & Park [22] investigated the effect of cooperation between pharmaceutical and bio companies and the direct and indirect effects of government R&D support on innovation performance. According to the research results, upstream partnerships were significantly related to companies' innovation performance, and government R&D support directly and indirectly affected companies' innovation by promoting internal R&D and domestic upstream and downstream cooperation. The importance of government R&D support and networking and cooperation between universities, research institutes, and subparts was emphasized. A study by Carayannopoulos & Auster [30] found that biopharmaceutical companies are more likely to source external knowledge through acquisition when the knowledge area is more complex and valuable when choosing acquisition or alliance when sourcing external knowledge, and that there is a higher possibility of sourcing external knowledge through alliance. The relationship between the two was also said to be strengthened.

Lin et al. [26] investigated the impact on innovation performance from the perspective of inter-firm R&D alliance experience and absorptive capacity as an essential mechanism for creating new technological knowledge. Firms with high absorptive capacity and firms with more alliance experience show more innovative performance, and in particular, innovation performance peaks when the technological distance from the alliance partner is at a medium level when interacted with the proportion of R&D alliances in the firm's alliance portfolio. did. In addition, it was said that R&D alliances complement rather than replace internal R&D within a company. A study by Xia & Roper [27] investigated the impact of the relationship between absorptive capacity and external relationships, two key aspects of open innovation, on the growth of small biopharmaceutical companies in the United States and Europe. Research results show that absorptive capacity plays an important role in a company's growth, and that exploratory relationships are largely dependent on the continuity of R&D in terms of a company's absorptive capacity and interaction with the outside world. On the other hand, participation in exploitative relationships is related to the company's absorptive capacity. It was said that there were more conditions regarding competency.

According to George et al. [28], pharmaceutical and bio companies' alliance portfolio characteristics and absorptive capacity together affect the company's innovative and financial performance. Lu et al. [29] classified them into inbound OI and outbound OI, respectively. The impact on a company's innovation performance was studied. Research results show that inbound and outbound OI have a positive effect on a company's innovation performance, and absorptive capacity positively regulates inbound and outbound OI and a company's innovation performance. As such, a variety of existing studies have been conducted on open innovation in biopharmaceutical companies, but the studies that have been conducted so far are fragmented and do not present integrated results. In addition, existing studies were mainly conducted in advanced countries in the biopharmaceutical industry, necessitating caution in interpreting the implications of open innovation in catching-up countries. In this regard, this study investigated how absorptive capacity, government R&D support, and alliance management capacity affect the moderating factors in the relationship between open innovation and performance, focusing on the Korean pharmaceutical and bio industry, a catch-up country. We aim to derive managerial and policy implications for open innovation in the biopharmaceutical industry by applying it to Korean companies, which are catching up in the biopharmaceutical industry, and analyzing and comparing them.

The remainder of this paper is organized as follows. Section 2 proposes hypotheses on technological innovation performance based on situational diversity. Section 3 introduces the data

and analysis methods and defines the variables. Section 4 presents and discusses the analysis results through a comparative analysis with other studies. Section 5 presents the implications for policy and management based on the results of the analysis.

## 2. Literature review and hypothesis

### 2.1 Open innovation in the biopharmaceutical industry

In the biopharmaceutical industry, corporate OI is no longer an option, but an essential strategic plan. Deloitte (2015) analyzed 281 global biopharmaceutical firms from 1999 to 2012 and revealed that the success rate of final new drug development for firms using an OI strategy was more than three times higher than that of firms using a closed innovation strategy. In particular, the biopharmaceutical industry is one of the most important fields in terms of value creation. It takes 12 to 15 years from basic research and development (R&D) through preclinical and clinical trials to commercialization, and can cost up to $800 million. Considering this [31], few firms have the financial and technical capabilities to participate in new drug development. This indicates that the strategy of securing the entire knowledge and technology required to develop new drugs within a firm is becoming difficult to implement [32]. OI is also important for the profits of biopharmaceutical firms, and commercializing this technology directly affects sales [33].

OI is when a firm appropriately utilizes inward and outward knowledge flows to accelerate internal innovation and expand the market for external utilization of innovation. OI involves not only producing and releasing technologies developed within a firm to the market, but also inbound open innovation, which involves developing technologies primarily developed within a firm into technologies that can be commercialized by external organizations. Knowledge and resources such as outbound OI, joint research, product development and commercialization, joint manufacturing, joint marketing, and joint ventures that absorb technology primarily developed by an external organization internally and develop it into a technology that can be commercialized. In exchange contracts, it can be separated and defined as a type of OI that combines inbound and outbound OI [7] Inbound OI involves exploring and leveraging technology and knowledge outside the enterprise and opens boundaries to access technological and scientific capabilities. Governance modes that provide inbound OI mechanisms to high-tech firms include in-licensing, acquisitions, joint ventures, and R&D contracts, and a representative example is selling technology as a method of outbound OI [34]. Dahlander and Gann [4] further divided the types of open innovation, classifying sourcing and acquisition as inbound and technology sales and disclosure as outbound OI [4].

Whether firms in the biopharmaceutical industry should adopt an OI strategy depends on the characteristics of biopharmaceutical technology [35]. First, biopharmaceutical technology is characterized by considerable uncertainty. Second, such technology has multidisciplinary characteristics. Third, it requires cumulative technology. Because technological uncertainty is high, biopharmaceutical firms proceed with one or two highly certain candidate substances in the pipeline and share the entire process from the discovery of candidate substances to conducting clinical trials step-by-step rather than solving them on their own. Furthermore, because biopharmaceutical firms are often based on one or two accumulated core technologies, complementary technologies from various fields must be integrated through OI to develop them into commercial technologies [36]. In other words, because biopharmaceutical technology requires capabilities accumulated through numerous failure processes, technologies in various fields cannot be developed simultaneously, which means that OI is inevitable.

In general, biopharmaceutical firms, mainly small- and medium-sized enterprises (SMEs), commercialize the basic research results of research institutions such as universities and

transfer them to pharmaceutical firms [34, 35]. This helps pharmaceutical firms commercialize the basic research results of research institutes and form alliances with other biopharmaceutical and pharmaceutical firms for technology development, manufacturing, marketing, and investment [22]. In particular, small start-ups lack the resources and capabilities to compete; therefore, they must strive to secure complementary assets through cooperation with various external organizations as a strategic choice [37]. Additionally, biopharmaceutical firms often do not start their businesses based on a variety of core technologies but rather do so with one or two accumulated core technologies. This means that innovation with other biopharmaceutical firms and the development of the industry require an organization capable of innovation [22].

## 2.2 Inbound open innovation and innovation performance in the biopharmaceutical industry

Inbound OI refers to the flow of knowledge from outside into the organization, which seeks to strengthen internal capabilities by introducing external technical resources and knowledge into the organization. In other words, an organization's technology exploration and knowledge acquisition and absorption capabilities become important. Specific forms include in-sourcing external knowledge, joint research, and venture investment. Inbound OI can lower the risk and cost of exploring new technologies through technology purchasing and improve a firm's flexibility, time to market, and NPD performance [38]. In addition, firms can solve these problems by purchasing technology licenses from external sources, that is, technology purchasing, as inbound OI can promote and accelerate the internal innovation process [39].

Technology purchasing is important for R&D-intensive (high-tech) firms because they have a high demand for innovation [40]. First, firms can attract R&D investment through technology purchases and utilize other organizations' resources and capabilities, including technology [36]. Additionally, technology purchases from competent organizations serve as positive signals that increase market value, which increases reputation and promotes additional technology purchases. Second, technology purchasing allows firms to integrate external ready technologies and leverage them to address gaps in the market, thereby advancing internal innovation processes [41]. Third, technology purchasing can help improve innovation performance while promoting the commercialization of technologies and enabling the development of complex products through the integration of proven technologies. Furthermore, it can improve innovation performance because it allows firms to develop complex products by integrating tested and proven technologies. Therefore, firms that utilize external resources can achieve greater success in creating innovation and generating sales. In fact, purchasing an already-developed technology can shorten the development time for new products and improve both innovation and financial performance by limiting preemptive monopoly and competitors' preemptive advantage in the market. However, purchasing extensive technology, that is, purchasing too much technology, may limit the development of important internal technical knowledge and reduce the potential for core technology development capabilities [17].

Biopharmaceutical firms can increase corporate innovation by acquiring external technology. The biopharmaceutical industry incurs enormous development costs because the product-development period from R&D to commercialization is relatively longer than that of other industries. These industrial characteristics have emphasized the need for firms to purchase technology from various research institutions such as universities, hospitals, government-funded research institutes, as well as pharmaceutical firms. Through technology purchasing, pharmaceutical firms have the advantages of reducing costs on internal development,

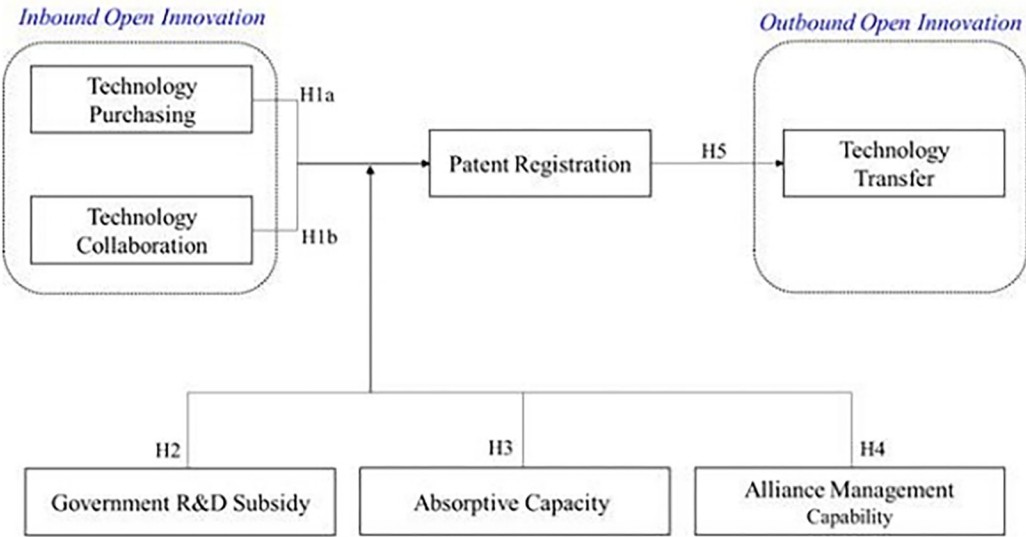

**Fig 1. Schematic diagram of research hypothesis.**

achieving rapid growth [41], and having access to cutting-edge technology. In addition, by selecting excellent technologies in advance, the risk of failure can be reduced, and through this, bio firms' R&D efficiency can be increased, and technological innovation performance can be promoted. Pharmaceutical firms thus increase their technological knowledge [24] and strengthen their technological capabilities [22] through external technology search and use processes. This leads to greater performance through product or process innovation. As depicted in Fig 1 this suggests our first hypothesis:

*Hypothesis 1a*: *Technology purchasing will have a positive effect on technological innovation of biopharmaceutical* firms

Technology collaboration between and among firms is becoming important in corporate innovation activities. As competition intensifies and technology advances rapidly in many industries, firms often need to develop new products more quickly and effectively. However, most are limited in size, have few internal resources, and a limited competency base [42]. Both technologies and products are becoming more complex, small businesses are finding it increasingly difficult to engage purely in product and technology development, and firms and institutions (universities) need the capabilities and knowledge required for such development. This requires cooperation with external partners [43], meaning that the role of OI is increasing for firms. Through OI activities, that is, collaboration, firms can provide access to scarce knowledge and technologies, reduce development costs, provide risk-sharing possibilities, and improve product-development processes [44].

Bianchi et al. [34] stated that both technology and products are becoming more complex, and it is increasingly difficult for firms to independently participate in product and technology development, so the role of open innovation is increasing. Because the capabilities and knowledge required for such development are dispersed across firms and institutions (universities), collaboration is done with external partners [32]. The most important issue in collaboration between firms is the uncertainty of partner collaboration. Strategic alliance is the accumulated experience of such collaboration. Among various open innovation methods such as strategic

alliance, joint venture, and M&A for R&D activities, strategic alliance is generally different from other methods. Considering the relatively low risks and costs compared to other methods, it is preferred by firms, and is the broadest form of collaboration that encompasses all forms of collaboration. Since strategic alliances are established for the strategic purposes of both firms promoting collaboration, they often involve one or more collaborations simultaneously in resource exchange, joint marketing, joint research and development, etc. [43]. In addition, by leveraging collaboration partners' external resources, firms can develop new technology combinations that allow them to explore wider markets, bridge internal technology gaps, test acquired technologies, and increase the speed and quality of innovation activities [39].

OI activities through knowledge sharing and collaboration are creating innovative performance for firms; OI basically means collaboration [45]. High-tech firms engage in extensive collaboration to secure external knowledge and accelerate technological innovation. Furthermore, as the capabilities and knowledge required for development are distributed across firms and institutions (universities), they collaborate with external partners. Collaboration increases the likelihood of goal achievement by securing additional resources and avoiding negative contingencies [46]. Such collaboration promotes innovation as a driving force for knowledge production and the creation of new innovations and expands a firm's knowledge base that can be exploited for knowledge redistribution or transfer.

Biopharmaceutical firms require cooperation with other firms for R&D. Many biopharmaceutical firms start with one or two specialized technologies and a pipeline [35]. The biopharmaceutical industry is also very complex, multidisciplinary, and utilizes technologies from a variety of fields [47]. The relatively long product-development period from R&D to commercialization incurs enormous development costs [23]. These characteristics have led to strategic alliances and collaborations not only with other biotechnology firms but also with various institutions such as universities, hospitals, and research institutes [23, 35]. Through this opportunity to acquire and learn complementary resources and capabilities from external organizations, biopharmaceutical firms can receive positive impacts, such as R&D performance and patent performance [23]. In addition, excellent scientific knowledge or basic technology from research institutes and biotechnology firms is transferred to pharmaceutical firms, which helps in terms of commercialization, improved financial performance, and technological innovation performance. However, cooperation with potential competitors can lead to technology leakage owing to exposure of core competencies, increased costs of finding and managing cooperation partners [36], and can encourage opportunistic behavior by partners. Accordingly, as depicted in Fig 1 this suggests our first hypothesis:

*Hypothesis 1b*: *Collaboration will have a positive effect on technological innovation in biopharmaceutical firms*

## 2.3 Controlling factors of open innovation

**2.3.1 Government R&D support.** Government R&D support for the biopharmaceutical industry plays an important role in a firm's technology investment and performance. The industry is technology-intensive. Therefore, biopharmaceutical firms should have higher R&D investments and intensity than firms in other industries. Although it is important for biopharmaceutical firms to make their own R&D investments, external financing, such as venture capital and debt financing, and public support, such as government R&D subsidies, play important roles. In addition, because the R&D process from discovery of candidate substances to clinical trials involves high risks owing to long R&D periods and large-scale R&D investments, firms' technological innovation performance needs to be supported through government R&D subsidies. Moreover, the industry needs government R&D support because,

approval of a new drug does not necessarily lead to a firm 's financial performance, and may incur significant manufacturing costs, and sales through distribution channels may not be easy. Government R&D subsidies promotes technology commercialization.

Government R&D subsidies either replace private firms' R&D or increase firms' R&D expenditures. A certain level of subsidy induces an increase in private R&D investment, whereas excessive subsidies displace it [47]. Guellec and Van Pottelsberghe De La Potterie [48] state that government R&D subsidies increase a firm's R&D investment but a crowding effect occurs when the subsidy exceeds 20% of such investment. Bérubé and Mohnen [49] established that firms that received government subsidies had 4% higher R&D intensity than those that did not. Czarnitzki and Licht [50] determined that firms that received government R&D subsidies had higher internal R&D than those that did not. Research has proven that more investment is being made in activities.

Government R&D subsidies generally positively affect technological innovation. Firms participating in R&D consortia and those that received government R&D subsidies were found to have higher patent performance than firms that did not receive subsidies [50]. Bérubé and Mohnen [49] proved that firms that received both tax benefits and government R&D subsidies generated more technological innovation results, such as NPD compared to those that received only tax benefits. Additionally, Kang and Park [22] demonstrated that government R&D subsidies for Korean biotechnology firms have direct and indirect positive effects on patent performance through R&D manpower, intensity, and alliances. Government R&D subsidies increase sales or profits by promoting technology transfer or product commercialization [51]. In other words, such subsidies promote technological innovation performance and technology transfer, which thereby increases corporate sales [52].

Meanwhile, government R&D subsidies above a certain level cause inefficiency and hinder organizational performance. Choi, et al. argued that government R&D subsidy support acts as a priming force to encourage firms' private investment up to a certain level, but excessive subsidies can cause moral hazard and hinder corporate innovation. Because complete monitoring of government R&D execution is impossible, problems such as moral hazard and inefficient execution of funds exist. Busom stated that the increase in government R&D subsidies limits firms' R&D input, and Czarnitzki and Licht [50] proved that R&D subsidies do not affect R&D and patents in Germany. Similar results were derived regarding Korea's government R&D subsidies and their effectiveness. Shin, et al. [53] examined the input, output, and behavioral additionality of Korean biotechnology firms through government R&D subsidies. In terms of input additionality, firms that received subsidies continued to increase their R&D investment for three years compared to those that did not compared to when the subsidies were first paid. In terms of output additionality, firms that received government R&D subsidies had higher technological innovation performance for the first one to two years than those that did not; however, little effect was observed after three years, and no evidence that financial performance was also higher was derived. From a behavioral additionality perspective, government R&D subsidies promoted strategic alliances and reduced firms' external financing and reduced the growth rate of debt financing compared to those that did not receive subsidies for three years after receiving government R&D subsidies. Government R&D subsidies have been proven effective in replacing corporate debt financing to some extent. As depicted in Fig 1, this suggests:

*Hypothesis 2*: *Government R&D support for biopharmaceutical firms will have a positive effect on regulating technological innovation performance.*

**2.3.2 Absorptive capacity.** The ability to evaluate and utilize external knowledge is important for technological innovation. Previous researchers argue that external knowledge can be

easily acquired, but Cohen and Levinthal [24] posit that external knowledge incurs costs to the recipient. In other words, resources must be invested to absorb external knowledge. Among the invested resources, internal R&D investment is the most important resource for creating new knowledge and absorptive capacity and indicates the extent to which a firm invests in NPD and R&D resources (e.g., human resources). Through internal R&D development, organizations can develop technological knowledge and better control and understand the tacit knowledge involved in the external technology acquisition process. Therefore, firms benefit from utilizing the new knowledge created through internal R&D investments to acquire technology from outside and achieve corporate performance [24].

*Firms* with high absorption of external knowledge investigate and make efforts to identify new technological opportunities. Thus, the ability to recognize the value of new external information, assimilate it, and apply it for commercial purposes is very important for innovation capability [24]. This capability is referred to as a firm's "absorptive capability." Cohen and Levinthal [24] stated that the importance of internal R&D for building absorptive capability is part of building prior knowledge and varies depending on the learning environment. Many studies have demonstrated that firms with high internal R&D have sufficient technology to recognize and assimilate external knowledge, and therefore require a high level of internal R&D to absorb such knowledge. Additionally, in the process of acquiring external knowledge, firms with high R&D intensity are more likely to recognize, assimilate, and utilize the value of new ideas, whereas those with low R&D intensity are less likely to develop superior technical knowledge capabilities [24]. In other words, if absorptive capacity is strong, technological innovation performance can be improved by minimizing conflicts between technologies acquired from external and internal organizations and maximizing complementary advantages.

Biopharmaceutical firms need the ability to assimilate, transform, and utilize the knowledge acquired from institutions according to their strategic goals [54]. In other words, biopharmaceutical firms must be able to absorb scientific knowledge from research institutes to achieve technological innovation results [55]. Firms can create synergistic innovation performance through absorptive capacity, that is, a type of dynamic capacity, and utilize this effect as a core competency [24, 54]. In the biopharmaceutical industry, strong technology-intensive absorptive capacity allows firms to improve technological innovation performance by minimizing conflicts between technologies acquired from external and internal organizations and maximizing complementary advantages. In addition, a firm's absorptive capacity plays an important role in the relationship between external OI and innovation performance [56]. In particular, to increase innovation performance, a strategy of starting OI in familiar fields (e.g., existing main research fields, existing therapeutic areas with existing strengths) and expanding opportunities for innovation is effective for performance generation [16].

However, low absorptive capacity prevents firms from generating technological innovation performance [15]. Firms with excellent absorptive capacity achieve efficient performance by accepting external knowledge, but their resources are limited. Interaction with the outside world requires significant resources and is expensive [21], so the cost aspect must be carefully considered. In particular, when absorptive capacity is insufficient, the cost of OI for technology consumers increases. People with low absorptive capacity have low learning ability and have a negative impact on the performance of technological innovation through collaboration [57–59]. Multifaceted reasons may exist behind this negative relationship, including firms' insufficient capacity to absorb knowledge and technology originating in other industries, or the resource drain created by the acquisition of external knowledge. Based on these considerations, we expect inbound OI to affect innovation and financial performance. Similarly, along with the positive effect of inbound OI on a firm's innovativeness, many studies have

documented that the external acquisition of knowledge negatively impacts a firm's innovation output [60, 61]. In addition, firms with low R&D intensity are less likely to develop good technical knowledge. Without sufficient knowledge capabilities, firms have fewer opportunities to recognize and understand the knowledge that underpins similarities and differences in partner technologies, which reduces innovation effectiveness. Therefore, as depicted in Fig 1, this suggests:

*Hypothesis 3*: *Absorptive capacity will have a positive effect on the technological innovation performance of biopharmaceutical firms.*

**2.3.3 Alliance management capability.** Firms enter into various R&D partnerships to secure external knowledge and accelerate technological innovation. Bigliardi and Galati [62] argue that factors such as lack of knowledge, collaboration experience, funds, and organization hinder OI in SMEs. Strategic alliances are voluntary agreements between firms to develop and commercialize new products, technologies, or services Gulati [63]. Alliance management is difficult because of the complexity and uncertainty of managing projects that cross organizational boundaries Kogut [57], Chesbrough [7] argued that the technological innovation process of firms engaged in high-tech industries is currently evolving from closed innovation to OI. The OI strategy of acquiring or transferring technology from external organizations plays an important role in creating technological innovation performance [8]. Among various OI methods such as strategic alliances, joint ventures, and M&A for R&D activities, strategic alliances are generally preferred by firms considering the relatively low risks and costs compared to other methods. Alliances for "learning" allows firms s to accelerate technological development by acquiring and utilizing knowledge developed by other firms [64]. Alliance management capability can be defined as how much alliance experience has been secured and accumulated. Alliance management capability is the ability of a firm to capture knowledge about alliance management, share and store this knowledge, and apply this knowledge to current and future alliances [65]. Firms secure and accumulate knowledge about alliance management by effectively utilizing alliance experience and converting it into knowledge [66]. Firms learn to manage through alliance experience and consequently develop Alliance management capability [67]. In other words, Alliance management capability is based on alliance management knowledge gained through experience with various partners, which is useful for future alliances regardless of partner type. Therefore, alliance management capabilities allow partners to adapt the types of information and knowledge, shared understanding, and common goals shared within the alliance to changes in the environment, thereby improving performance [65]. The positive association between recent alliance experience and performance reflects the importance of dynamic capabilities.

Alliance management capabilities must be built through accumulated alliance experience, and firms with more alliance experience should be able to manage more alliances productively. The alliance management function is a dependent capability built over time through repeated participation in strategic alliances, and the ability to effectively manage alliances is a dynamic capability that can integrate a firm. To respond to a rapidly changing environment, firms create innovative forms of competitive advantage by building and reorganizing internal and external capabilities. Firms with excellent alliance management capabilities are more effective in competing with other firms and can gain the upper hand. A firm's experience in managing alliances has a positive effect on patent rates, NPD, and stock market value creation. Therefore, strategic alliances have become an important key to corporate success in high-tech industries. Various types of strategic alliances can lead to different technological innovation outcomes in the biopharmaceutical industry. Strategic alliances are important for biopharmaceutical firms

in terms of corporate growth [23, 68]. The chances of survival increase through the results created through strategic alliances between such firms, and the survival rate can be increased through alliances on commercialization, such as R&D, manufacturing, and marketing [23, 28, 40, 68]. This is particularly important for firms in high-tech industries, which rely on extensive inter-firm collaboration for the discovery, development, and commercialization of new products.

However, a lack of alliance management ability can hinder innovation and creation by making it difficult to quickly optimize the innovation process. In general, OI performance is assumed to increase as the number of collaborations within an organization increases; however, when it exceeds a certain level, the complexity of OI transactions increases, and performance actually decreases. Therefore, organizations must find an appropriate amount of cooperation, and the existing degree of OI (quality and number of cooperation) of the collaborating organization also becomes an important factor in innovation performance. To improve innovation performance through OI, organizations must increase their initial alliance experience and accumulate and develop capabilities. Previous alliance experience can improve the performance of OI by increasing the organization's absorptive capacity; in the case of organizations with little alliance experience, the creation of innovative performance in OI may be insufficient owing to increased uncertainty and transaction costs [47]. Purdy, et al. [31] view this alliance experience as the ability to manage alliances and document that if alliance management capability is lacking, it is difficult to quickly optimize the innovation process, which can hinder the creation of innovation performance. In particular, it may initially seem beneficial for biopharmaceutical firms to open their corporate boundaries to external knowledge and technology and access new markets, but this may not be true (negative) in the highly competitive advanced biopharmaceutical field (biopharmaceuticals) [31]. These results indicate that in an era of advanced technologies, volatile environments, and strong competition, firms cannot avoid the limitations of OI; however, understanding OI strategies will help mitigate such impacts. Therefore, as depicted in Fig 1, this suggests:

*Hypothesis 4*: *Alliance management capabilities will have a positive effect on the technological innovation performance of biopharmaceutical firms.*

## 2.4 Outbound open innovation and innovation performance in the biopharmaceutical industry

Technology transfer is an important strategy for outbound OI. As the product life cycle is shortened, the period of profit generation through new products and services is becoming shorter. To generate profits in terms of NPD competition, firms commercialize internal assets through external organizations [7] and intentionally leak knowledge to expand the market. Outbound OI refers to earning profits by putting ideas on the market, selling patents, and transferring ideas to the external environment [69], and includes technology sales, licensing, and spin-offs. Firms can have financial and strategic advantages in outbound OI, leverage technological knowledge outside their boundaries, or co-develop it with other organizations.

Technology transfer is a type of partnership between firms that is especially important in technology-intensive industries. Technology transfer is the process of transferring or diffusing knowledge and technology from one person or firm to another entity [65, 70]. This process involves further development and use of the acquired technology in new applications, materials, products, processes, or services. This involves efforts to share knowledge, skills, and processes among various actors. Technology transfer generally occurs among universities, governments, and corporations [71], and decoupling capabilities must be improved in such

OI. The decoupling capability is a firm's ability to identify and transfer knowledge for external use and consists of external identification and external commercialization [72]. External identification is a firm's ability to recognize external technology transfer opportunities [72], and external commercialization is the transfer of internal technology to the outside [15], which enables firms with insufficient resources to commercialize new technologies. This process involves licensing unused assets and selling them to outside parties.

Technology transfer is an essential component of the biopharmaceutical industry. Biopharmaceutical firms seek to create added value through technology transfer because they lack the financial and material resources to commercialize technology [73]. Technology transfer plays an important role as a source of various innovations in the R&D process of the industry [32] as it lowers the total cost and risk of new drug development, and shortens the time to market. Technology transfer between pharmaceutical and research-intensive bio firms is actively underway. Depending on the various R&D processes from drug discovery and development, biopharmaceutical firms utilize internal ideas and technologies, integrate (M&A) with various external organizations (e.g., universities or competing firms) to exchange technology and knowledge, or increase partnerships. The business model has changed, including the reorganization of R&D [74]. As a result, biotech and small pharmaceutical firms carry out internal ideas, technologies, and R&D projects and transfer technology to large pharmaceutical firms. The latter increase sales by commercializing technology and R&D projects, whereas the former supplement their finances by selling products and technologies [74]. As Fig 1 suggests:

*Hypothesis 5*: *Technological innovation of biopharmaceutical firms will have a positive effect on technology transfer.*

The above figure structures the study's hypotheses in Fig 1. This study investigates the relationship between inbound open innovation (technology purchasing and technology collaboration) and innovation performance (patents) of biopharmaceutical firms, and what moderating factors are related between innovation performance (patents) and outbound open innovation (technology transfer). Technology purchasing is important for R&D-intensive (high-tech) firms because they have a high demand for innovation [40]. Firms can attract R&D investment through technology purchases and utilize other organizations' resources and capabilities, including technology [36], It can develop internal innovation processes by integrating and utilizing external prepared technologies to address market gaps [41]. Technology purchasing can improve sales generation and financial performance by shortening the time for new product development and creating innovation outcomes [38], as it allows companies to develop complex products through the integration of proven technologies [39].

This requires cooperation with external partners [43], meaning that the role of OI is increasing for firms. Through OI activities, that is, collaboration, firms can provide access to scarce knowledge and technologies, reduce development costs, provide risk-sharing possibilities, and improve product-development processes [44]. Collaboration represents a distinct type of open innovation because it involves mutual innovation activities with common goals and the active participation of external stakeholders [69]. High-tech firms engage in extensive collaboration to secure external knowledge and accelerate technological innovation. Furthermore, as the capabilities and knowledge required for development are distributed across firms and institutions (universities), they collaborate with external partners. Collaboration increases the likelihood of goal achievement by securing additional resources and avoiding negative contingencies [46]. Such collaboration promotes innovation as a driving force for knowledge production and the creation of new innovations and expands a firm's knowledge base that can be exploited for knowledge redistribution or transfer.

In particular, in the biopharmaceutical industry, open innovation is no longer an option but an essential strategic measure. Pisano [13] finds the reason in the characteristics of biopharmaceutical [13]. First, biopharmaceutical has high uncertainty, second, it has multidisciplinary characteristics, and third, it requires technological accumulation. Because of the high uncertainty of technology, biopharmaceutical firms proceed with one or two highly certain candidate substances in the pipeline, and also do not solve all processes from candidate substance discovery to clinical trial performance on their own, but rather choose to divide the work by stage. Furthermore, since biopharmaceutical firms often base their efforts on one or two accumulated core technologies, they must integrate complementary technologies from various fields through open innovation in order to develop them into commercial technologies [40]. In other words, because pharmaceutical and biotechnology requires accumulated capabilities through numerous failures, it is impossible to develop technologies from various fields together, which means that open innovation is inevitable. These industrial characteristics provide pharmaceutical and biotechnology firms with strategic advantages such as avoiding high costs for internal development, achieving rapid growth [41], and accessing cutting-edge technologies through technology purchases and collaborations with various research institutes such as universities, hospitals, and government-funded research institutes as well as pharmaceutical firms [22, 24]. Therefore, biopharmaceutical firms can select excellent technologies from biotechnology firms in advance, reduce the risk of failure during development, and thereby increase the R&D efficiency of biotechnology firms, thereby promoting technological innovation results.

In addition, through this opportunity to acquire and learn complementary resources and capabilities from external organizations, biopharmaceutical firms can receive positive impacts, such as R&D performance and patent performance [23]. In addition, excellent scientific knowledge or basic technology from research institutes and biotechnology firms is transferred to pharmaceutical firms, which helps in terms of commercialization, improved financial performance, and technological innovation performance. It is schematized. In addition, this study hypothesizes the moderating effects of government R&D support, absorptive capacity, and alliance management capacity on the relationship between inbound open innovation and innovation performance, and the relationship between innovation performance and outbound open innovation (technology transfer) as a moderating factor. set. Since innovation depends on a firm's ability to make external linkages and manage the innovation process [7], we propose that two specific types of organizational capabilities, namely alliance management capabilities and absorptive capabilities, will affect innovation performance. The government's R&D support serves as a source of funds for initial technology investment by firms with insufficient funds, promotes external cooperation or financing, and indirectly strengthens the company's R&D alliance by strengthening the company's absorptive capacity [75, 76]. On the other hand, companies that receive government R&D support have a crowding-out effect in which firms own R&D investment is replaced by government R&D support, showing a negative relationship [77]. Additionally, there is a possibility of moral hazard in using government R&D support for purposes other than research and development. There are also studies that show that there is no significant relationship between government R&D support and innovation performance. Therefore, efforts are needed to break away from the mixed results between government support and innovation performance and find better outcome variables that can verify the effectiveness of government R&D support. Additionally, strengthening a firm's absorptive capacity increases the possibility of strategic alliances with various organizations [78]. Therefore, this study investigated how absorptive capacity, government R&D support, and alliance management capacity influence the moderating factors in the relationship between open innovation and performance, focusing on the Korean biopharmaceutical industry, a catch-up country.

## 3. Methodology

### 3.1 Data

The data used in this study were obtained from biopharmaceutical firm information from the small and medium-sized venture database provided by the bio-innovation linkage service of the National Biotechnology Policy Research Center (BioIN) under the Korea Research Institute of Bioscience and Biotechnology.

Firms are classified into small- and medium-sized based on sales. Firms first select a list based on several criteria such as region, firm size, and initial public offering (IPO), and then build data on their OI activities. Since there was no database, data on joint research, research agreements, technology purchases and sales, and spin-offs were constructed by referring to a Naver news search, the target firm's website, and business reports. Basic information on Korean biopharmaceutical firms is provided by KoDATA (KOREA RATING & DATA), a big data platform organization in the financial industry that specializes in corporate credit research and evaluation and provides credit rating information.

Patent data were collected from the patent database of the Korea Patent Information Service, and the number and amount of government R&D were collected from the government R&D support database of the National Science and Technology Knowledge Information Service. Data were constructed by referring to other accessible data as much as possible. In Korea, there is no site that provides all information about biopharmaceutical firm information. Corporate information (sales, firm size, etc.) was collected from KoDATA (KOREA RATING & DATA), information related to open innovation was collected from the website, Naver, and business reports, patent information was collected from the Korea Patent Information Service, and the number and amount of government R&D cases were collected from National Science & Technology Information Service.

Firms for which it was difficult to find detailed information in Korean firm data or were outside the normal scope of the biopharmaceutical industry were excluded. The initial sample comprised 2798 biopharmaceutical firms, and 688 firms in the industry were selected according to bioindustry classification, excluding the biofood, biomedical device, biochemical/energy, and bioenvironmental industries. The goal of this study was limited to the biopharmaceutical industry, and heterogeneous characteristics were excluded in order to report only on biopharmaceutical firms.

Additionally, 161 firms that were closed, firms with missing or inaccurate information, and simple wholesale and retail firms that did not conduct R&D were excluded.

Information on biopharmaceutical firms includes corporate types classified as sole proprietors, general corporations, foreign firms, KONEX, and KOSDAQ, and includes small-molecule drugs, biopharmaceuticals, new-concept treatments, animal drugs, element technology development, biosensors, in vitro diagnostics, pharmaceutical raw materials, and so on. It was classified as Red Bio as a material. Biopharmaceutical c firms were classified according to the value chain of research, development, production, and sales. Additionally, we determined the year of establishment of the firm and recorded the firms' age. Furthermore, the location was classified by region, and employees were surveyed. Data on the 527 biopharmaceutical firms used in this study are presented to supporting information (S1 Data).

### 3.2 Variables

The number of registered patents was used as the dependent variable. Owing to technological innovation, where securing patent rights is important, some biopharmaceutical firms receive high praise despite not having a specific profit model. Most of these firms tend to have

excellent patent rights. Additionally, many pharmaceutical patents protect related products, which reflects their importance. Pharmaceutical patents are also important in preparing for future patent infringements because they allow for individual patents, not only for drug candidates and organisms themselves, but also for manufacturing processes and related technologies. Multiple patents for the production of a single finished product are also allowed. Therefore, we used the number of registered patents to measure the technological innovation performance of biopharmaceutical firms.

The types of inbound and outbound OI according to the type of OI were considered as independent variables. First, biopharmaceutical technology has great uncertainty, second, it has multidisciplinary characteristics, and third, it requires cumulative technology [35]. Firms with these characteristics develop technologies primarily developed within the firm into technologies that can be commercialized by external organizations (inbound OI) and absorb technologies primarily developed by external organizations internally. OI, which includes developing technologies that can be commercialized (outbound OI), is increasing. Additionally, inbound and outbound OI may have complementary characteristics to some extent. To determine whether the effect of outbound OI on corporate performance is strengthened or weakened as the level of inbound OI increases, technology purchase and cooperation corresponding to inbound OI and technology transfer corresponding to outbound OI were used as independent variables. Collaboration was based on domestic and foreign joint research in the relevant year, and strategic partnership was defined and used as the cumulative number of collaboration research. Firms secure and accumulate knowledge about alliance management by effectively utilizing and accumulating alliance experience [66]. Through alliance experience, they learn how to manage and consequently develop alliance management capability [67]. In other words, the more strategic experience, the more trackable the management capabilities are, so the cumulative number of strategic alliances was used as a proxy for alliance management capabilities. Therefore, we investigated the impact of inbound and outbound OI on corporate performance and whether the moderating factors moderate the effect of OI.

Firm size, business experience, venture certification, research institute ownership, and diversification were used as control variables. The number of employees was used to measure firm size, and the firm's age was calculated by subtracting the year of establishment from 2022, the point in time of the data, along with firm size. This is the most commonly used variable in the literature. Generally, a firm's size and age have a proportional relationship with its technological innovation performance. The larger the firm, the more resources it devotes to technological innovation, and the older the firm, the greater its accumulated knowledge capabilities. However, some studies document that as the size and age increase, organizational inertia also increases, which may have a negative impact on the firm's technological innovation performance. Therefore, we controlled for firm size and age. Venture certification and IPO are variables that were not easily found in the existing literature. In particular, venture certification is a system unique to Korea that selects SMEs with potential. The standards for certification are similar to the selection criteria for government support in many ways, so they are expected to affect whether to receive government support. Owning a research institute indicates whether a firm has its own affiliated research institute or a dedicated department at the organizational level.

R&D intensity was used as a proxy variable for absorptive capacity to test the effect of controlling the absorptive capacity of biopharmaceutical firms. Internal R&D investments not only create new knowledge, but also contribute to a firm's absorptive capacity, allowing it to access, transform, and use new types of knowledge [24]. Many studies argue that effective inbound OI requires a higher level of internal R&D because such firms have sufficient relevant technological knowledge to recognize and assimilate external knowledge [19]. During the

**Table 1. Variables and definitions.**

| Variable | | Operational Definition |
|---|---|---|
| Dependent Variables | *PATENT* | Number of patents registered with the Korean Intellectual Property Organization |
| | *TRANSFER* | Number of technology transfer from other domestic and foreign firms |
| Independent Variables | *PURCHASE* | Number of technology purchases from other domestic and foreign firms |
| | *COLLABO* | Number of research collaboration with Other domestic and foreign Institutions |
| Moderating Variables | GOV_SUP | Number of government R&D subsidies |
| | ABSORP_CAP | R&D expenses to revenues |
| | *ALLIANCE* | Accumulated number of strategic alliances |
| Control Variable | *SIZE* | Number of employees |
| | *AGE* | Number of years since founding |
| | *VENTURE* | 1 if the firm underwent Venture Certification, 0 otherwise |
| | RND_CTR | 1 if there is an R&D center in the firm, 0 otherwise |
| | *DIVERSIFIC* | Number of business areas(research, development, manufacturing, marketing, cmo, cro) in which the firm is engaged |

inbound OI process, firms with high R&D intensity generally have well-developed technological knowledge and are more likely to recognize the value of new ideas, facilitate the assimilation of new technological knowledge, and exploit external opportunities. Therefore, we considered internal R&D investment, or R&D intensity, as a controlling factor and investigated how it affects innovation performance. In addition, the government's R&D support serves as a source of funds for initial technology investment by firms with insufficient funds, promotes external cooperation and financing, and indirectly strengthens the firm's R&D alliance by strengthening its absorptive capacity.

Additionally, owing to the lack of venture capital, the Korean biotechnology industry is inherently reliant on government support. Among the various effects of government support, government R&D funds have been evaluated as having a direct impact on the technological innovation performance of biotechnology firms. Additionally, a series of studies have concluded that government R&D funds indirectly contribute to technological innovation performance by promoting R&D input and strategic alliances for R&D activities [22]. To verify the moderating effect of absorptive capacity of biopharmaceutical firms, R&D intensity was used as a proxy variable for absorptive capacity. It was calculated by dividing research and development costs by sales [79]. As a proxy for government R&D funds, the number of government R&D projects was used. There are many firms that carry out more than one government-supported R&D project per year, and the amount of funding may vary depending on the project in the data. For alliance management capabilities, the cumulative number of collaborations was used as an experience indicator of how many alliances there were. Therefore, we used absorptive capacity, government R&D support, and alliance management capacity as control variables in the relationship between OI and performance, focusing on the Korean biopharmaceutical industry, a catch-up country. The definitions of the variables are presented in Table 1.

### 3.3 Analysis method

The number of registered patents, which is a dependent variable, is also a count variable, a non-negative integer that has multiple zeros. To utilize count data, the Poisson distribution or the negative binomial distribution must be considered in the analysis [80]. When using the Poisson model, the average and the variation of dependent variables should correspond to each other. According to previous studies, many count data display over—dispersion, meaning that the dispersion exceeds the average [81]. Over-dispersion is considered to be caused by unobserved heterogeneity, and as a frequent solution to this problem, it is assumed that the

parameter of the Poisson distribution moves according to a specific probability distribution [82]. In this study, the average VIF was 3.64, as a result of the multicollinearity test.

The dependent variable, the number of registered patents, is a count variable and is a non-negative integer with multiple zeros. To use count data, a negative binomial distribution must be considered in the analysis. Additionally, because patent registration takes time, we assumed a time lag of one to two years [22]. A negative binomial regression was used in Phase 1 of this study. We adopted the following model for analysis. In addition, to evaluate the causal effect of technological innovation performance (patents) on technology transfer, which is outbound innovation, we used two-stage least squares (2SLS) to account for potential endogeneity. After obtaining the predicted values by regressing the endogenous explanatory variables on all the exogenous variables, including the instrumental variables, we replaced these with the predicted values obtained in the first step. 2SLS can handle multiple endogenous explanatory variables and multiple instruments.

In this study, negative binomial regression and Poisson binomial regression tests were performed, and it was determined that negative binomial regression analysis was an appropriate analysis model because the log likelihood value (-2063.3818), AIC (Akaike Information Criterion) value (4164.764), and BIC (Bayes Information Criterion) value (4261.851) were smaller than those of the Poisson analysis results. The detailed methods and results are as follows.

The Poisson regression rarely fits in practice since in most applications the conditional variance is greater than the conditional mean. If the mean structure is correct, but inefficient. Further, the standard errors from the Poisson regression model will be biased downward, resulting in spuriously large z-value.

In this study, Negative binomial regression and Poisson binomial regression tests were performed to obtain information on the goodness of fit of the two stages of the IV regression. The negative binomial regression results showed that the Log likelihood value was –2063.3818, the AIC was 4164.764, and the BIC was 4261.851. The Poisson binomial regression results showed that the Log likelihood value was –2084.8472, the AIC (Akaike Information Criterion) was 5783.694, and the BIC (Bayes Information Criterion) was 5870.562.

Also, the likelihood-ratio test is a test for the overdispersion parameter alpha (our results show that the Likelihood-ratio test of alpha = 0: chibar2(01) = 1558.42 Prob> = chibar2 = 0.000). When the overdispersion parameter is 0, the negative binomial distribution is identical to the Poisson distribution. In this case, alpha is significantly different from 0, so we emphasize once again that the Poisson distribution is not appropriate. In other words, Testing for Overdispersion (a test to statistically justify why Negative binomial regression vs Poisson binomial regression model is used) rejected Ho within a statistically significant range, so NBR was used rather than the Poisson regression model.

AIC (Akaike Information Criterion) and BIC (Bayes Information Criterion) are standard measures for model selection, and a regression equation with a smaller value, whether it is the AIC value or the BIC value, is a more appropriate regression equation. In other words, a smaller AIC and BIC means that the model has the largest degree of coupling (likelihood) and the smallest number of variables. In this study, considering the analysis results, negative binomial regression was judged to be an appropriate analysis model.

In addition, after performing 2SLS, an estimation method using instrumental variables (IV) was implemented, and the F test statistic was less than 10 (F test statistic = 3.72096), so it was judged that the correlation with the endogenous variable was weak. The detailed explanation and method are as follows.

In this study, an estimation method using instrumental variables (IV) was implemented after performing 2SLS. Instrumental variables must not necessarily be correlated with the error term of the regression model and must be correlated with the endogenous explanatory

variables. In IV estimation, the number of instrumental variables must be greater than or equal to the number of endogenous explanatory variables, and when there are more instrumental variables than endogenous explanatory variables, it is called over-identification. In the case of over-identification, it is necessary to test the validity of the instrumental variables, that is, to conduct an over-identification test. Therefore, in this study, an over-identification test was additionally conducted after 2SLS. If the F test statistic is greater than 10, it can be judged that there is a correlation with the endogenous variables. In the results of our study, the F test statistic = 3.72096, indicating a weak correlation. The Sagan and Basmann p-values ($p = 0.0916$, $p = 0.0938$) of the over-identification test are greater than 0.05, so the instrumental variables are not correlated with the error term at the 5% significance level.

According to previous research, firms promote technological innovation performance through technology purchases and technological cooperation from various organizations. OI does not necessarily lead to positive innovation performance, which can be influenced by government R&D support, absorptive capacity, and alliance management capacity. Technological innovation performance is also affected by firm size, age, venture certification, ownership of a research institute, and diversification. Therefore, reflecting the results of previous studies, firm size, age, venture certification, and possession of a research institute were included and controlled as control variables, and government R&D support, absorptive capacity, and alliance management capacity were used as control variables.

A regression model was set up to analyze the impact of technological innovation performance on a firm's technology transfer performance. In addition, the control variables that affect technology transfer performance were reflected in the second stage. In the first step, the endogenous variables were regressed, and predicted values were obtained; in the second step, the variables were replaced with the predicted values in the equations of interest and estimation.

## 4. Results and discussion

The basic statistics and correlations of the variables and the negative binomial regression and 2SLS results are presented in Tables 2 and 3, respectively. Technology purchasing did not have a positive effect on technological innovation performance, but technological cooperation exhibited a positive relationship ($p<0.01$) (Hypothesis 1a was not supported; Hypothesis 1b was supported). Firms increase their external technical knowledge and strengthen their technical capabilities through technology purchases. The extent to which the acquisition of external

**Table 2. Descriptive statistics and correlations between variables.**

| | Mean | Std. Dev. | Min | Max | 1 | 2 | 3 | 4 | 5 | 6 | 7 | 8 | 9 | 10 | 11 | 12 |
|---|---|---|---|---|---|---|---|---|---|---|---|---|---|---|---|---|
| *PATENT* | 2.035 | 4.237 | 0 | 48 | 1 | | | | | | | | | | | |
| *TECH-TRANS* | 0.087 | 0.473 | 0 | 9 | 0.163 | 1 | | | | | | | | | | |
| *PURCHASE* | 0.103 | 0.465 | 0 | 7 | 0.147 | 0.228 | 1 | | | | | | | | | |
| *COLLABO* | 0.322 | 0.923 | 0 | 10 | 0.236 | 0.245 | 0.239* | 1 | | | | | | | | |
| *ABSORP_CAP* | 6.131 | 18.207 | 0 | 187 | 0.456 | 0.337 | 0.236* | 0.280* | 1 | | | | | | | |
| *GOV_SUP* | 0.984 | 1.621 | 0 | 12 | 0.330 | 0159 | 0.110* | 0.265* | 0.203* | 1 | | | | | | |
| *ALLIANCE* | 0.262 | 1.164 | 0 | 15 | 0.055 | 0.185 | 0.225* | 0.227* | 0.223* | 0.103* | 1 | | | | | |
| *SIZE* | 62.651 | 81.073 | 0 | 1165 | 0.119 | 0.114 | 0.097* | 0.170* | 0.201* | 0.157* | 0.154* | 1 | | | | |
| *AGE* | 27.511 | 19.453 | 7 | 131 | 0.179 | 0.212 | 0.225* | 0.245* | 0.443* | 0.191* | 0.311* | 0.188* | 1 | | | |
| *VENTURE* | 0.500 | 0.500 | 0 | 1 | 0.127 | 0.005 | -0.022 | 0.041* | -0.175* | 0.178* | -0.078* | -0.142 | -0.269* | 1 | | |
| *RND_CTR* | 0.813 | 0.405 | 0 | 5 | 0.245 | 0.117 | 0.130* | 0.188* | 0.117* | 0.255* | 0.113* | 0.012* | 0.179* | 0.555* | 1 | |
| *DIVERSIFIC* | 2.059 | 1.232 | 1 | 4 | 0.222 | 0.192 | 0.224* | 0.241* | 0.369* | 0.214* | 0.292* | 0.344* | 0.534* | -0.128* | 0.280* | 1 |

**Table 3. Results of 2SLS regression (N = 527).**

| | Model 1 | | Model 2 | | Model 3 | | Model 4 | |
|---|---|---|---|---|---|---|---|---|
| | *PATENT* | *TRANSFER* | *PATENT* | *TRANSFER* | *PATENT* | *TRANSFER* | *PATENT* | *TRANSFER* |
| *PATENT* | | 0.055*** (0.005) | | 0.074*** (0.009) | | 0.087*** (0.012) | | 0.048*** (0.005) |
| *PURCHASE* | 0.017 (0.086) | | 0.127* (0.075) | | 0.100* (0.061) | | 0.051 (0.100) | |
| *COLLABO* | 0.247*** (0.042) | | 0.253*** (0.044) | | 0.219*** (0.034) | | 0.268*** (0.057) | |
| *ABSORP_CAP* | 0.021*** (0.002) | | | | | | 0.018*** (0.002) | |
| *GOV_SUP* | | | 0.184*** (0.015) | | | | 0.167*** (0.017) | |
| *ALLIANCE* | | | | | 0.017 (0.048) | | -0.064** (0.049) | |
| *PURCHASE × ABSORP_CAP* | 0.001 (0.001) | | | | | | 0.002** (0.001) | |
| *COLLABO × ABRORP_CAP* | -0.005*** (0.001) | | | | | | -0.004*** (0.001) | |
| *PURCHASE × GOV_SUP* | | | -0.029* (0.022) | | | | -0.034** (0.025) | |
| *COLLABO × GOV_SUP* | | | -0.030*** (0.010) | | | | -0.030* (0.012) | |
| *PURCHASE × ALLIANCE* | | | | | -0.013(0.034) | | -0.018 (0.034) | |
| *COLLABO × ALLIANCE* | | | | | -0.002(0.034) | | 0.029** (0.041) | |
| *SIZE* | 0.001 (0.001) | 0.001 (0.001) | 0.001* (0.001) | 0.001 (0.001) | 0.001*** (0.001) | 0.001 (0.001) | 0.001** (0.001) | 0.001 (0.001) |
| *AGE* | 0.001 (0.002) | 0.003*** (0.001) | 0.002** (0.002) | 0.002** (0.001) | 0.005* (0.002) | 0.002* (0.001) | -0.001** (0.002) | 0.003*** (0.00) |
| *VENTURE* | 0.616*** (0.096) | 0.015 (0.030) | 0.472*** (0.084) | -0.001 (0.028) | 0.609*** (0.086) | -0.009 (0.029) | 0.463*** (0.096) | 0.018 (0.030) |
| *RND_CTR* | 0.385*** (0.098) | -0.019 (0.034) | 0.479*** (0.076) | -0.037 (0.032) | 0.464*** (0.081) | -0.048 (0.034) | 0.381*** (0.093) | -0.015 (0.034) |
| *DIVERSIFIC* | 0.111*** (0.039) | 0.003 (0.012) | 0.121*** (0.033) | 0.004 (0.012) | 0.114*** (0.034) | 0.001 (0.012) | 0.130*** (0.038) | 0.005 (0.013) |

Notes

***, **, * represents $p<0.01$, $p<0.05$, $p<0.1$, respectively. Standard errors are between brackets next to the coefficient.

technological knowledge affects a firm's performance depends on its internal R&D efforts, and Cohen and Levinthal [24] stated that a firm's internal development expertise improves the firm's absorptive capacity. In other words, firms with a high level of internal R&D can better integrate technologies acquired externally and achieve higher performance. Technical cooperation involves conducting joint research, etc. with partners, which means that a firm has a network partner. Becoming proficient in applying OI in cooperation with others contributes to the creation of innovative performance by compensating for a lack of internal resources and capabilities [57, 58].

R&D intensity and government R&D support are positively related to technological innovation performance (all $p<0.01$). Government R&D subsidies have a positive effect on new product development and firms' patent performance (technological innovation performance). R&D intensity is defined as the internal organization's willingness to conduct research and development, that is, the ratio of total internal R&D expenditures to sales. In other words, through internal R&D investment, conflicts between technologies acquired from external organizations and the internal organization's technologies are minimized and complementary

advantages are maximized to influence technological innovation performance. Technology-intensive industries should have higher R&D intensity than other industries. Firm's must have high R&D of their own, and the greater the external financing, the more positive the impact. Technological innovation performance is important in technology-intensive industries. Government R&D support for the biopharmaceutical industry plays an important role in a firm's technology investment and performance.

This means that government subsidies positively contribute to the creation of technological innovation performance by biopharmaceutical firms in the Korean biopharmaceutical industry, which suffers from a lack of private funding [57]. Thus, government R&D funds positively contributes to the competitiveness of firms. This result supports Cefis and Marsili's [83] study that technological innovation positively contributes to performance, while Blanes and Busom, [84] argue that government R&D subsidies are provided to firms with excellent R&D capabilities, intensity, and performance. In addition, the "Picking the Winner" principle, which states that firms with excellent R&D capabilities and performance benefit from government R&D subsidies, has been proven effective in the Korean biotechnology industry [54]. Pal [1] suggested that R&D intensity has a positive impact on Indian pharmaceutical firms' business sustainability. Alliance management capabilities had a negative effect on technological innovation performance ($p<0.05$, $p<0.01$, separate). This indicates that the performance of strategic alliances can be influenced by contingency factors depending on the characteristics of the partners, which may also differ because the purpose and function of each alliance relationship is [54]. This is contrary to the common belief that the more experience in collaboration, the more active the collaboration with various entities. This means that even if there is a lot of experience in cooperation, it does not necessarily lead to technological innovation results. The biopharmaceutical industry incurs enormous development costs because the product development period from R&D to commercialization is relatively long compared to other industries [85]. Due to these industrial characteristics, strategic alliances between firms are increasing with various organizations such as universities, hospitals, government-funded research institutes, and biotechnology firms as well as pharmaceutical firms [13, 17]. The performance of these strategic alliances may differ because each type of alliance partner has different alliance motivations, resources and capabilities, organizational structure and culture, and degree of competition with partners [53]. Pharmaceutical and biotechnology firms and biotechnology firms acquire cutting-edge scientific knowledge through cooperation with research institutes and universities, and research institutes create scientific knowledge that can be used in industry [46, 47]. In other words, technological innovation performance can be improved by acquiring and learning professional knowledge through interaction with excellent researchers at research institutions and universities. On the other hand, partnerships between biotechnology firms and pharmaceutical firms can maintain R&D investments from pharmaceutical firms and utilize the pharmaceutical firm's resources and capabilities, such as facilities, personnel, and technology. These advantages improve technological innovation performance by promoting technology commercialization of biotechnology firms [9]. In addition, alliances between biotechnology firms are formed to create new technological innovation results by integrating the different technologies each possesses [16]. For this reason, high-tech firms are forming various alliances to secure external knowledge and accelerate technological innovation [55], but because the purpose and function of each alliance is different, contingency factors may be different [45] results are consistent with the results.

Technology purchase and cooperation, adjusted by R&D intensity, had a negative moderating effect on technological innovation performance ($p<0.05$, $p<0.01$). This implies that absorptive capacity as a proxy for R&D intensity activates the limits of inbound OI for technological innovation (Hypothesis 3 is not supported). A firm's ability to internalize external

knowledge can influence the extent to which it can achieve higher innovation performance through collaboration; this ability depends on the firm's internal capabilities, such as in-house R&D, production experience, and technical training. In addition, regarding the moderating effect of R&D intensity on the relationship between a firm's technological cooperation and innovation performance, R&D intensity plays an important role in a firm's use of external knowledge [83]. This is because a firm cannot benefit from external knowledge flows simply by being exposed to external knowledge, but instead develops internal technological capabilities that can recognize the value of new external knowledge and then assimilates and utilizes it. In other words, the more external knowledge flows in, the more important the role of absorptive capacity is in securing competitive advantage.

Government R&D subsidy had a positive effect on companies' technological innovation performance. However, the government's R&D support, which is a moderating factor, had a negative moderating effect on technology purchase and technological cooperation ($p < 0.05$, $p < 0.1$, separate). Thus, additionality through public support, as a proxy for government R&D support, activates the limits of inbound OI on technological innovation performance (Hypothesis 2 is not supported).

There is no positive effect on technological innovation performance adjusted for alliance management ability, but technological cooperation adjusted for alliance management capability has a negative effect on technological innovation performance ($p < 0.05$). This indicates that alliance management, as a proxy for alliance experience, activates the limits of technology purchases for technological innovation and the limits of technological collaboration for technological innovation performance (Hypothesis 4 is not supported). This result contradicts previous findings that strategic alliances enhance biotechnology firms' chances of survival by complementing their resources and capabilities and providing them with an advantage in managing external competition or institutional challenges [23]. Alliance management is difficult because of the complexity and uncertainty inherent in managing projects that cross organizational boundaries [86]. Rothaermel and Deeds [40] found that alliance management capabilities are the most important link between alliances and NPD in high-tech ventures. There is an inverted U-shaped relationship, which means that after a certain point, total returns decrease. This is because it is limited by the firm's past investments, experience, and resources the firm currently possesses. Similarly, firm-level capabilities are limited, including the number of alliances a firm can manage, and performance declines when a firm's activities exceed its finite capabilities [40]. Strategic alliances are a possible alternative to securing essential resources outside a firm's boundaries, and alliance management is an important strategic area that allows organizations to change their resource bases. However, despite the proliferation of strategic partnerships, they do not necessarily lead to results. Empirical evidence shows that approximately 50% of alliances fail to meet expectations [87]. Anand and Khanna [66] find that alliance performance varies considerably from firm to firm. Although some firms may gain significant benefits from alliances, others may experience failures [88]. Thus, a firm's alliance management capabilities play an important role in explaining why some firms have higher alliance success rates than others.

Using the 2SLS method, we examine whether technological innovation performance affects outbound innovation (technology transfer). In the first stage, we examine the impact of inbound OI (technology purchasing, technology cooperation) and regulating factors (R&D intensity, government R&D support, and alliance management capacity) on technological innovation performance. In the second stage, we examine outbound OI (technology transfer). When examining these effects, all were positive ($p < 0.01$, $p < 0.05$, $p < 0.1$, all). External technology acquisition refers to the absorption of external technological knowledge, for example, through licensing deals or strategic alliances [89, 90], whereas external technology exploitation

refers to the exclusive commercialization of technological knowledge or internal technology [33]. According to Arora, et al. [91], the size of a firm's technology portfolio affects the scope of external technology transfer and its patent status affects the scope of technology transfer. In addition, Grindley and Teece [92] stated that patents are essential facilitators of technology transactions, making technology transactions possible. This is consistent with the findings of Grindley and Teece [92] and Lichtenthaler [93], who found that by licensing or selling technology through these patents, firms actively try to realize financial and strategic opportunities to commercialize technology (Hypothesis 5 is supported).

Firm size and venture certification had a positive effect on technological innovation performance ($p < 0.05$, $p < 0.01$, separate), while firm age had a negative effect on technological innovation performance ($p < 0.05$). Therefore, large corporations have an advantage in achieving technological innovation results, and in the case of the venture certification system, the certification system of the Korean government promotes the innovation activities of start-up firms. Certification as an innovative firm provides various benefits and is positive for business and innovation performance. In other words, firms that have been comprehensively evaluated by existing corporate evaluation agencies are mainly selected so that they can be considered trustworthy.

The existence and diversification of an R&D organization had a positive effect on technological innovation performance ($p < 0.01$, all). Dedicated R&D department contributes to the development of the ability to understand and predict the technological needs of key firms. Additionally, rather than being a repository of a firm's core technological capabilities from which internal innovation opportunities are generated and pursued, R&D departments perform an important intermediary function to effectively utilize external technological sources [94]. Once external technological knowledge is absorbed, the R&D department acts as an internal hub for synthesizing, reorganizing, and aligning knowledge related to various technological areas and originating from external and internal sources [65].

## 5. Conclusions

Based on a sample from the biopharmaceutical industry in Korea, a catching-up country, from 2014 to 2021, this study uses negative binomial analysis and 2SLS to determine how absorptive capacity, government R&D support, and alliance management capacity affect the relationship between OI and performance. This was estimated using the stepwise least squares method. Existing studies have mainly been conducted in the biopharmaceutical industry in advanced countries. Therefore, the relationship between performance creation and regulating factors in Korea's unique biopharmaceutical industry has not been developed, and studies to date have not focused on OI and innovation performance. Furthermore, fragmentary results have been derived. Therefore, our empirical results reveal that technology purchase does not have a positive effect on technological innovation performance in biopharmaceutical firms' inbound OI; however, technological cooperation has a positive effect on technological innovation performance [10, 11]. In addition, technology purchasing cooperation, moderated by government R&D support, absorptive capacity, and alliance management capacity, exerted a negative adjustment effect on technological innovation performance but a positive effect on technology transfer performance. Thus, OI does not necessarily lead to positive innovation performance but is mixed with negative ones, and regulating factors (government R&D support, absorptive capacity, and alliance management capacity) activate the limitations.

This study has academic and practical implications for the biopharmaceutical industry. We quantitatively demonstrated that, as hypothesized, OI does not always lead to positive effects as in previous studies, and that it is controlled by regulating factors (government R&D

support, absorptive capacity, and alliance management capacity). This is a result of the constraints on Korea's biopharmaceutical industry. OI cannot result in innovation performance given the situational characteristics of Korea's biopharmaceutical industry. Innovation performance should be considered by comparing and analyzing cases. The main limitations of OI are activated by government R&D support, the ability to absorb external knowledge (technology), and accumulated partnership experience. Thus, OI must be appropriately utilized to create technological innovation. This study expands the method of investigating the limitations of OI by applying the inbound OI strategy to Korean biopharmaceutical firms, and can be compared and analyzed with developed countries by applying the same to other catching-up countries in the future.

Managers of biopharmaceutical firms investigate and make efforts to identify new technological opportunities in firms with a high absorption of knowledge (technology) outside the organization according to the type of OI presented in the study. Thus, the ability to recognize the value of, assimilate, and apply new external information for commercial purposes is important for innovation capability [24]. Many studies document that firms with high internal R&D require a high level of internal R&D to absorb external knowledge (technology) [37]. When alliance management capabilities are heterogeneously distributed among firms and difficult to imitate, such capabilities have the potential to create a competitive advantage at the corporate level [84]. We propose that firm-level alliance management capabilities may be particularly salient for high-tech entrepreneurial firms. Given the importance of resource access for new ventures, these firms often need to rely on extensive inter-firm collaboration in the discovery, development, and commercialization of new products. Successful NPD is particularly important for entrepreneurs in high-technology industries. Firms must build the capacity to commercialize technological innovations created from inbound OI and generate profits by linking them to technology transfer or commercialization. This can provide managers of biopharmaceutical firms with strategic guidance for OI. Managers can improve corporate performance by reviewing the diversity factors that can affect OI, such as the type of open innovation presented herein, resources (including knowledge resources) or those that need to be supplemented, and the firm's alliance management capabilities. This can provide strategic help in making comprehensive decisions.

Because government R&D subsidies have a negative impact on technological performance, policymakers need to prepare policies so that firms can invest more in OI for technological innovation without the side effects of R&D support. When selecting firms for government R&D support, not only technical aspects, but also the size and age of firms in countries that are developing in the biopharmaceutical industry should be considered. Previous research indicates that firm size and age are positively related to government R&D support. If government R&D subsidies are mainly provided to young, small, and medium-sized firms, such as startups with high technological innovation potential or capabilities, these firms may find it difficult to overcome financial constraints in sustainable technology development. In particular, because biopharmaceutical firms are based on one or two accumulated core technologies, the size and age of the firm, its technological innovation potential and capabilities, and so on, need to be considered in the process of selecting support targets. Because government R&D subsidies are based on preventing the contraction of R&D activities owing to technological imperfections and the risk of corporate technology development and commercialization, government R&D subsidies are naturally granted to firms with high technological innovation. should be selected and supported in consideration of potential or ability.

Additionally, government R&D subsidies are needed for biopharmaceutical firms to strengthen their commercialization capabilities. Previous research has determined that government R&D subsidies increase corporate sales by promoting technological innovation

performance and technology transfer [41], but the present results do not support this. This means that Korean biopharmaceutical firms require government R&D subsidies to strengthen their innovation performance. The new drug development process proceeds in the following order: basic exploration and original technology research process, development candidate selection stage, preclinical (non-clinical) test stage, clinical trial process, and new drug approval and marketing. In addition, the process, which carries high levels of risk, takes an average of 10 years and costs more than 1.2 trillion won, and the average success rate from pre-clinical to final commercialization is 9.6. As such, the high risk continues until the commercialization stage, and must be expanded to the commercialization stage [39]. In this context, it is highly likely that government R&D subsidies for biopharmaceutical firms will remain at the basic R&D stage and will not be able to continue the technological innovation performance and commercialization stages. Therefore, the government must promote firms' financial performance through active government R&D support for commercialization.

For latecomers, government R&D subsidies play an important role in the growth and survival of small and medium-sized businesses. This is a solution to the lack of private capital, such as venture capital, for latecomer companies in government-led catch-up countries such as Korea, China, and India compared to developed countries such as the United States. Paradoxically, this requires Asian latecomers to increase private investment. Various types of strategic alliances can lead to various technological innovation outcomes in the biopharmaceutical industry. For biopharmaceutical companies, strategic alliances are important in relation to the growth of the company. The alliance management function is a dependent capability built over time through repeated participation in strategic alliances. In order to respond to a rapidly changing environment, companies create innovative forms of competitive advantage by building and reorganizing internal and external capabilities. Companies with excellent alliance management capabilities can gain an advantage over other companies. In order to develop the biopharmaceutical industry in catching-up countries, strategic alliances must be further activated, and catching-up countries in the biopharmaceutical industry must encourage strategic alliances between companies to form an industrial ecosystem that can provide complementary assets to companies. Pisano [13] argued for the importance of strategic alliances, referring to biopharmaceutical companies with unique core competencies in countries with advanced biopharmaceutical industries. Additionally, the results of this study show that the absorptive capacity of pharmaceutical and bio companies must be increased in order to acquire and learn implicit scientific knowledge, and the difference in the absorptive capacity of the two organizations may have a negative impact on the learning process of external organizations. Meanwhile, it has been proven that strengthening the internal capabilities of pharmaceutical and bio companies through R&D investment is of utmost importance for successful cooperation. Regarding the degree of open innovation, Chesbrough [32] placed equal importance on internal knowledge base and external technology and paradoxically emphasized the importance of internal technological capabilities. In the same context, this study showed that when a company's absorptive capacity is insufficient or acts as a regulating factor in the performance of open innovation, it can be toxic to the company. These results suggest that in an era of advanced technologies, volatile environments, and strong competitive forces, biopharma companies in catch-up countries cannot avoid the dark side of OI, but understanding OI sourcing strategies will help mitigate such impacts.

Although this study has academic and practical implications, it has the following limitations. First, owing to a lack of data, we used the number of employees between 2014 and 2021 as the firm size. Furthermore, the number of employees was used under the assumption that there was no significant difference from the size of the firm; however, it was used on the premise that size may vary depending on the growth of the firm. Therefore, more complete data are

required for future studies. Additionally, this study is limited in that it does not consider the time lag between innovation activities and performance because it focuses on one- and short-term data. Therefore, follow-up studies should use survey data covering a longer period. Second, a further limitation of this study is the lack of generalization of the findings beyond the context of the data examined. Therefore, these results may not be applicable to all industries as they mainly reflect inbound OI, technology purchasing, and collaboration. Additionally, owing to the highly regulated and high-tech nature of biopharmaceutical firms and their business sectors, complete generalizations cannot be made about all firms. Moreover, the outbound OI indicator was analyzed by applying technology transfer; however, the impact on innovation performance should be examined from various perspectives, including various indicators, and this result should be studied in other fields (i.e., low technology, etc.). In addition, out of the 2,798 firms surveyed, only 527 pharmaceutical and bio firms and firms with complete information were used, which was a limitation. Future research is needed to supplement the data.

## Supporting information

**S1 Data.**
(CSV)

## Author Contributions

**Conceptualization:** Changhyeon Song, Kwangsoo Shin.

**Data curation:** HyeJoo Wang, Changhyeon Song, Kwangsoo Shin.

**Formal analysis:** HyeJoo Wang.

**Methodology:** Changhyeon Song, Kwangsoo Shin.

**Project administration:** Kwangsoo Shin.

**Supervision:** Kwangsoo Shin.

**Validation:** Kwangsoo Shin.

**Writing – original draft:** HyeJoo Wang, Kwangsoo Shin.

**Writing – review & editing:** Kwangsoo Shin.

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
