## [Decision Letter · Decision Letter 0]

9 Apr 2024

PONE-D-23-42382Does the impact of open innovation depend on contextual factors? A case of the Korean biopharmaceutical industryPLOS ONE

Dear Dr. Shin,

Thank you for submitting your manuscript to PLOS ONE. After careful consideration, we feel that it has merit but does not fully meet PLOS ONE’s publication criteria as it currently stands. Therefore, we invite you to submit a revised version of the manuscript that addresses the points raised during the review process.

Dear Author,

I recommend to fully implement all the referee's suggestions. The manuscript has potential but needs a very significant improvement to be considered for publications. One of the reviewer is more positive, while the other suggests for rejection. 

Best regards.

We look forward to receiving your revised manuscript.

Kind regards,

Eleonora Pierucci

Academic Editor

PLOS ONE

Journal Requirements:

Reviewers' comments:

Reviewer's Responses to Questions

**Comments to the Author**

1. Is the manuscript technically sound, and do the data support the conclusions?

Reviewer #1: Yes

Reviewer #2: No

2. Has the statistical analysis been performed appropriately and rigorously? 

Reviewer #1: Yes

Reviewer #2: No

3. Have the authors made all data underlying the findings in their manuscript fully available?

Reviewer #1: No

Reviewer #2: No

4. Is the manuscript presented in an intelligible fashion and written in standard English?

Reviewer #1: Yes

Reviewer #2: Yes

5. Review Comments to the Author

Reviewer #1: Dear author/s, I’ve read this paper titled “Does the impact of open innovation depend on contextual factors? A case of the Korean biopharmaceutical industry” with interest. I think it has a good potential, but several amendments are needed to improve its quality and justify its publication. Below are my comments, good luck.

-The background and purpose of the study should be mentioned in the Introduction. Currently, there is a lack of logical basis for addressing contextual factors (Government R&D support, Absorptive capacity, Alliance management capability) in the impact of open innovation. And more existing literature on the moderating effects of these factors should be added in Introduction section.

-In Literature review part, open innovation (OI) is divided into ‘inbound’ and ‘outbound’ OI. The explanation for this needs to be supplemented.

-In Table 1, please describe the exact operational definition of DIVERSIFIC variable in detail.

-There is an error in the results of model 3 in Table3. The moderating effects of ALLIANCE with independent variables are missing. Please correct this.

-In Conclusion part, supplement the conclusion by focusing on the implications to the latecomers in biopharmaceutical industry. And the text of this part should be streamlined with contents derived from analysis results.

-Lastly, the content of this manuscript makes sense, but the English sentences need further correction. Especially, the abstract is an important part of deciding whether to read the paper. I strongly recommend proofread the whole sentences.

Reviewer #2: The paper is discretely written, and its aim is well presented. The work on the literature review and the link to the existing literature is sound and intriguing. Also the focus on South Korean firms is a strength of the analysis.

However several concerns arise when evaluating the empirical section which is pivotal as the main findings rely on the applied work. Here are my main comments:

- The definition of Open Innovation, is presented only in page 4. Throughout the introduction no mention is reported on what the author refer to when talking about OI (for example in page 2 there is a reference on the fact that “OI is a broad concept defined in various ways” but no explication is reported).

- Page 2, last paragraph: “Few studies have examined the relationship between an integrated perspective and the moderating factors in the impact of OI on technological innovation performance, and the results are inconsistent.” No work is mentioned to support such statement.

- It is not clear in the text what is the difference between collaborations (one of the main covariates) and strategic alliances (one of the ‘mediating variables’). “Strategic alliances are voluntary agreements between companies to develop and commercialize new products, technologies, or services” while – on the other hand – “This requires cooperation with external

- partners [26], meaning that the role of OI is increasing for companies. Through OI activities, that is, collaboration, companies can provide access to scarce knowledge and technologies, reduce development costs, provide risk-sharing possibilities, and improve product-development processes [27]”. There is need to provide more detail of the differences between the two and why you think you are capturing different things. Moreover, how the collaboration influences the cumulated stock of collaboration that should be linked to the strategic alliance management?

- The schematic diagram of research hypothesis need additional comment as it appears to be disjointed from the work.

- Some details on how data has been collected are missing. For instance:

o What does it mean that “Data were constructed by referring to other accessible data as much as possible”?

o What is the reason for excluding the biofood, biomedical device, biochemical/energy, and bioenvironmental industries?

o How passing from an initial number of 2798 companies to 527 might influence your results. Moreover, among these firms, how many are included in the regressions that you show in the paper? As you should know, no information on the number of observations included in the regressions is provided to the reader.

o The proxies for the variables included in the empirical work are not commented. They are reported only in the table 1.

- The results section is not sound.

o A table with descriptive statistics of all the variables is missing (also the dependent ones and information on minimum and maximum values).

o “Technology purchasing did not have a positive effect on technological innovation performance, but technological cooperation exhibited a positive relationship (p<0.01), which indicates that the company has network partners” this period does not make sense.

o “R&D intensity and government R&D support are positively related to technological innovation performance (all p<0.01), indicating that the biopharmaceutical industry is technology intensive” does not make sense either.

o p.19, incomplete sentence: “This indicates that the performance of strategic alliances can be influenced by contingency factors depending on the characteristics of the partners, which may also differ because the purpose and function of each alliance relationship is [38].”

o The empirical exercise is not complete. No information on the number of observations or the goodness of fit is provided. No support to the need to adopt a negative binomial (e.g. showing a graph of the distribution of the patents) is provided. No additional statistics on the test of endogeneity of the instrumented variable is provided. It is not clear why inward OI variables – adopted in the first stage – are not adopted also in the second stage as they might provide insights on the technology transfer. Also a test on overidentification of the instruments would have been useful in order to appreciate the appropriateness of the empirical analysis.

o It is not clear why the purchase*alliance and collabo*alliance are included only in the model 4 and not – as one would expect coherently with models 1 and 2 – also in model 3. In general, to understand the existence of a ‘moderating’ effect of absorp_cap, gov_sup and alliance one would require to see also the effect of the estimates with only purchase and collabo variables.

o Some concerns on the interpretation of the results arise as well. For instance, in page 19 the author writes that The results support the findings of Zhang and Guan [63] and Shin, et al. [37], proving that government R&D support does not have a positive effect or may have a negative effect on a company’s technological innovation performance. However, the impact of gov. support on technological innovation is positive (model 2 and 4). What is negative is the interaction with the variables purchase and collabo. This indicates that, although positively associated with the dependent variable, government support is concave with respect to purchase and collabo and vice versa purchase and collabo are concave with respect to government support.

6. PLOS authors have the option to publish the peer review history of their article (what does this mean?). If published, this will include your full peer review and any attached files.

Reviewer #1: No

Reviewer #2: No

---

## [Author Response · Author response to Decision Letter 0]

23 May 2024

**Response to Reviewer #1 Comments

Thank you very much for all your comments. We are very glad that you give us the opportunity to revise our manuscript. We have worked hard to address your excellent and detailed comments. Based on them, we have substantially improved this manuscript. We hope you like the changes we made.

Point 1: The background and purpose of the study should be mentioned in the Introduction. Currently, there is a lack of logical basis for addressing contextual factors (Government R&D support, Absorptive capacity, Alliance management capacity) in the impact of open innovation. And more existing literature on the moderating effects of these factors should be added in Introduction section.

Response 1: Thank you for your insightful comment. As you pointed out, the submitted manuscript lacked an explanation of research background and purpose in Introduction part. An explanation of the three variables (Government R&D support, Absorptive capacity, and Alliance management capacity) used as moderating effect variables was added, as well as information deriving the purpose of the study.

Modification #1 (line 106 – line 134 in revised manuscript)

Firm must invest resources to absorb knowledge outflow. Among invested resources, internal R&D investment has been considered the most important resource for creating new knowledge and absorptive capacity. Internal R&D investment refers to the extent to which a firm invests in internal activities (e.g., new product development) and research and development resources, and internal R&D investment is said to be important in developing technological knowledge that contributes to absorptive capacity [19]. Firms benefit by utilizing new knowledge created through internal R&D investments to acquire technology from outside and achieve corporate performance [19]. Effective inbound OI requires a higher level of internal R&D because firms with a high level of internal R&D have sufficient relevant technological knowledge to recognize and assimilate external knowledge [20]. Inbound OI suggests that firms with high R&D intensity generally have well-developed technological knowledge, which makes them more likely to recognize the value of new ideas, facilitate the assimilation of new technological knowledge, and exploit external opportunities. In contrast, firms with low R&D intensity are less likely to develop good technical knowledge capabilities [21]. Alliance management is a difficult organizational activity due to the complexity and uncertainty inherent in managing projects that cross organizational boundaries [22], and alliance management capability is a path-dependent capability that is built over time through repeated participation in strategic alliances, a firm's partnership experience has a positive impact on patent rates, new product development, and stock market value creation [7]. In addition, when alliance management capabilities are heterogeneously distributed among firms and difficult to imitate, a firm's alliance management capabilities have the potential to create a competitive advantage at the corporate level [23], and there is an inverse relationship between alliances in high-tech ventures and new product development. It showed a U shape. In this way, there are systematic differences in the alliance management capabilities of recent firms, and these differences can be a source of competitive advantage at the firm level [24]. Therefore, understanding how alliance-specific and firm-level factors influence a firm's ability to manage alliances is an important question, especially in the entrepreneurial context. Government R&D support are an important element of technological innovation and a key factor contributing to a sustainable economy. Failure to properly invest in R&D by private firms increases its importance, and implementing a technological innovation strategy to accelerate R&D has become one of the government's most important industrial policies. Because R&D is usually accompanied by market failures, government funding can increase R&D activity and move it closer to the social optimum. Additionally, R&D intensity and subsidies are positively correlated with the likelihood of innovation [2]. Government-supported R&D creates additional innovation for beneficiaries, but government R&D support above a certain level cause in efficiencies and hinder organizational performance.

Point 2: In Literature Review part, open innovation (OI) is divided into ‘inbound’ and ‘outbound’ OI. The explanation for this needs to be supplemented.

Response 2: Thank you for your insightful comment. As requested, we described the concept of OI by dividing it into ‘inbound’ and ‘outbound’ prior to the main text of Literature Review. We hope this will be helpful in understanding the paper.

Modification #1 (line 145 – line 163 in revised manuscript)

In particular, the biopharmaceutical industry is one of the most important fields in terms of value creation. It takes 12 to 15 years from basic research and development (R&D) through preclinical and clinical trials to commercialization, and can cost up to $800 million. Considering this [50], few firms have the financial and technical capabilities to participate in new drug development. This indicates that the strategy of securing the entire knowledge and technology required to develop new drugs within a firm is becoming difficult to implement [56]. OI is also important for the profits of biopharmaceutical firms, and commercializing this technology directly affects sales [69].

OI is when a firm appropriately utilizes inward and outward knowledge flows to accelerate internal innovation and expand the market for external utilization of innovation. OI involves not only producing and releasing technologies developed within a firm to the market, but also inbound open innovation, which involves developing technologies primarily developed within a firm into technologies that can be commercialized by external organizations. Knowledge and resources such as outbound OI, joint research, product development and commercialization, joint manufacturing, joint marketing, and joint ventures that absorb technology primarily developed by an external organization internally and develop it into a technology that can be commercialized. In exchange contracts, it can be separated and defined as a type of OI that combines inbound and outbound OI [13]. Inbound OI involves exploring and leveraging technology and knowledge outside the enterprise and opens boundaries to access technological and scientific capabilities. Governance modes that provide inbound OI mechanisms to high-tech firms include in-licensing, acquisitions, joint ventures, and R&D contracts, and a representative example is selling technology as a method of outbound OI [16]. Dahlander and Gann (2010) further divided the types of open innovation, classifying sourcing and acquisition as inbound and technology sales and disclosure as outbound OI [4].

Point 3: In Table 1, please describe the exact operational definition of ‘DIVERSIFIC’ variable in detail.

Response 3: Thank you for your valuable comment. As suggested, we modified the operational definition of ‘DIVERSIFIC’ variable in more detail. Please confirm the modification below.

Modification #1 (line 550 in revised manuscript)

Table 1. Variables and definitions

Variable Operational Definition

Dependent

Variables PATENT Number of patents registered with the Korean Intellectual Property Organization

 TRANSFER Number of technology transfer from other domestic and foreign companies

Independent

Variables PURCHASE Number of technology purchases from other domestic and foreign companies

 COLLABO Number of research collaboration with Other domestic and foreign Institutions

Moderating

Variables GOV_SUP Number of government R&D subsidies

 ABSORP_CAP R&D expenses to revenues

 ALLIANCE Accumulated number of strategic alliances

Control

Variable SIZE Number of employees

 AGE Number of years since founding

 VENTURE 1 if the firm underwent Venture Certification, 0 otherwise

 RND_CTR 1 if there is an R&D center in the firm, 0 otherwise

 DIVERSIFIC Number of business areas(research, development, manufacturing, marketing, cmo, cro) in which the firm is engaged

Point 4: There is an error in the results of model 3 in Table 3. The moderating effects of ALLIANCE with independent variables are missing. Please correct this.

Response 4: Thank you for your thorough review. The part you mentioned was omitted in the process of tabulating the analysis results.

Modification #1 (line 703 in revised manuscript)

Table 3. Results of 2SLS regression

 Model 1 Model 2 Model 3 Model 4

 PATENT TRANSFER PATENT TRANSFER PATENT TRANSFER PATENT TRANSFER

PATENT 0.055*** (0.005) 0.074*** (0.009) 0.087*** (0.012) 0.048*** (0.005)

PURCHASE 0.017 (0.086) 0.127* (0.075) 0.100* (0.061) 0.051 (0.100) 

COLLABO 0.247*** (0.042) 0.253*** (0.044) 0.219*** (0.034) 0.268*** (0.057) 

ABSORP_CAP 0.021*** (0.002) 0.018*** (0.002) 

GOV_SUP 0.184*** (0.015) 0.167*** (0.017) 

ALLIANCE 0.017 (0.048) -0.064** (0.049) 

PURCHASE × ABSORP_CAP 0.001 (0.001) 0.002** (0.001) 

COLLABO × ABRORP_CAP -0.005*** (0.001) -0.004*** (0.001) 

PURCHASE × GOV_SUP -0.029* (0.022) -0.034** (0.025) 

COLLABO × GOV_SUP -0.030*** (0.010) -0.030* (0.012) 

PURCHASE × ALLIANCE -0.013(0.034) -0.018 (0.034) 

COLLABO × ALLIANCE -0.002(0.034) 0.029** (0.041) 

SIZE 0.001 (0.001) 0.001 (0.001) 0.001* (0.001) 0.001 (0.001) 0.001*** (0.001) 0.001 (0.001) 0.001** (0.001) 0.001 (0.001)

AGE 0.001 (0.002) 0.003*** (0.001) 0.002** (0.002) 0.002** (0.001) 0.005* (0.002) 0.002* (0.001) -0.001** (0.002) 0.003*** (0.00)

VENTURE 0.616*** (0.096) 0.015 (0.030) 0.472*** (0.084) -0.001 (0.028) 0.609*** (0.086) -0.009 (0.029) 0.463*** (0.096) 0.018 (0.030)

RND_CTR 0.385*** (0.098) -0.019 (0.034) 0.479*** (0.076) -0.037 (0.032) 0.464*** (0.081) -0.048 (0.034) 0.381*** (0.093) -0.015 (0.034)

DIVERSIFIC 0.111*** (0.039) 0.003 (0.012) 0.121*** (0.033) 0.004 (0.012) 0.114*** (0.034) 0.001 (0.012) 0.130*** (0.038) 0.005 (0.013)

Notes: ***, **, * represents p<0.01, p<0.05, p<0.1, respectively. Standard errors are between brackets next to the coefficient. 

Point 5: In Conclusion part, supplement the conclusion by focusing on the implications to the latecomers in biopharmaceutical industry. And the text of this part should be streamlined with contents derived from analysis results.

Response 5: Thank you for your insightful comment. As you pointed out, we have supplemented the implications by focusing on latecomers. In addition, implications were derived by relating them to the results of existing literature.

Modification #1 (line 769 – line 792 in revised manuscript)

For latecomers, government R&D subsidies play an important role in the growth and survival of small and medium-sized businesses. This is a solution to the lack of private capital, such as venture capital, for latecomer companies in government-led catch-up countries such as Korea, China, and India compared to developed countries such as the United States. Paradoxically, this requires Asian latecomers to increase private investment. Various types of strategic alliances can lead to various technological innovation outcomes in the biopharmaceutical industry. For biopharmaceutical companies, strategic alliances are important in relation to the growth of the company. The alliance management function is a dependent capability built over time through repeated participation in strategic alliances. In order to respond to a rapidly changing environment, companies create innovative forms of competitive advantage by building and reorganizing internal and external capabilities. Companies with excellent alliance management capabilities can gain an advantage over other companies. In order to develop the biopharmaceutical industry in catching-up countries, strategic alliances must be further activated, and catching-up countries in the biopharmaceutical industry must encourage strategic alliances between companies to form an industrial ecosystem that can provide complementary assets to companies. Pisano [14] argued for the importance of strategic alliances, referring to biopharmaceutical companies with unique core competencies in countries with advanced biopharmaceutical industries. Additionally, the results of this study show that the absorptive capacity of pharmaceutical and bio companies must be increased in order to acquire and learn implicit scientific knowledge, and the difference in the absorptive capacity of the two organizations may have a negative impact on the learning process of external organizations. Meanwhile, it has been proven that strengthening the internal capabilities of pharmaceutical and bio companies through R&D investment is of utmost importance for successful cooperation. Regarding the degree of open innovation, Chesbrough [26] placed equal importance on internal knowledge base and external technology and paradoxically emphasized the importance of internal technological capabilities. In the same context, this study showed that when a company's absorptive capacity is insufficient or acts as a regulating factor in the performance of open innovation, it can be toxic to the company. These results suggest that in an era of advanced technologies, volatile environments, and strong competitive forces, biopharma companies in catch-up countries cannot avoid the dark side of OI, but understanding OI sourcing strategies will help mitigate such impacts.

Point 6: Lastly, the content of this manuscript makes sense, but the English sentences need further correction. Especially, the abstract is an important part of deciding whether to read the paper. I strongly recommend proofread the whole sentences.

Response 6: In the case of the draft, the expressions were somewhat awkward because they did not undergo separate English proofreading. In addition to the comments you pointed out, we reviewed and revised the English expressions throughout the manuscript. Thank you.

**Response to Reviewer #2 Comments

Thank you very much for all your comments. We are very glad that you give us the opportunity to revise our manuscript. We have worked hard to address your excellent and detailed comments. Based on them, we have substantially improved this manuscript. We hope you like the changes we made.

Point 1: The definition of Open Innovation, is presented only in page 4. Throughout the introduction no mention is reported on what the author refer to when talking about OI (for example in page 2 there is a reference on the fact that “OI is a broad concept defined in various ways” but no explication is reported)

Response 1: Thank you for your comment. As you pointed out, the exact definition of the concept of open innovation was missing although this paper deals with open innovation. A detailed explanation of OI concept has been added to the manuscript.

Modification #1 (line 73 – line 93 in revised manuscript)

Open innovation means intentionally allowing the inflow and outflow of knowledge into a firm to utilize external knowledge in value proposition design through a decentralized rather than centralized innovation process [7]. By including the financial and non-monetary benefits that can accrue to a variety of stakeholders, researchers are increasingly recognizing open innovation as a value co-creation process whose benefits extend beyond the enterprise [8]. Open innovation supports the establishment of a distributed innovation system in which companies open their internal innovation processes to external knowledge and technology [7, 8]. Unlike closed innovation systems, it also supports extending a firm's knowledge search strategy beyond its boundaries [9]. In this way, companies engage with customers [10], suppliers [11]and non-governmental organizations [11], involving various stakeholders, such as competitors, in a value creation strategy [8]. To integrate these stakeholders, companies can build a variety of engagement strategies across a variety of co-creation events and processes, such as crowdsourcing [12].

Open innovation consists of three forms: inbound, outbound, and combined [13]. The inbound open innovation process invites various extern

---

## [Decision Letter · Decision Letter 1]

15 Jul 2024

PONE-D-23-42382R1Does the impact of open innovation depend on contextual factors? A case of the Korean biopharmaceutical industryPLOS ONE

Dear Dr. Shin,

Thank you for submitting your manuscript to PLOS ONE. After careful consideration, we feel that it has merit but does not fully meet PLOS ONE’s publication criteria as it currently stands. Therefore, we invite you to submit a revised version of the manuscript that addresses the points raised during the review process.

We look forward to receiving your revised manuscript.

Kind regards,

Eleonora Pierucci

Academic Editor

PLOS ONE

Additional Editor Comments:

Dear Authors,

we received the report of the reviewers. One is satisfied, while the second, although appreciating the effort of the first round, still asks for some improvements. I kindly ask you to take into consideration all the comments carefully and revise the manuscript accordingly.

Reviewers' comments:

Reviewer's Responses to Questions

**Comments to the Author**

1. If the authors have adequately addressed your comments raised in a previous round of review and you feel that this manuscript is now acceptable for publication, you may indicate that here to bypass the “Comments to the Author” section, enter your conflict of interest statement in the “Confidential to Editor” section, and submit your "Accept" recommendation.

Reviewer #1: All comments have been addressed

Reviewer #2: (No Response)

2. Is the manuscript technically sound, and do the data support the conclusions?

Reviewer #1: Yes

Reviewer #2: Partly

3. Has the statistical analysis been performed appropriately and rigorously? 

Reviewer #1: Yes

Reviewer #2: No

4. Have the authors made all data underlying the findings in their manuscript fully available?

Reviewer #1: Yes

Reviewer #2: Yes

5. Is the manuscript presented in an intelligible fashion and written in standard English?

Reviewer #1: Yes

Reviewer #2: Yes

6. Review Comments to the Author

Reviewer #1: I have carefully reviewed the revised manuscript submitted by the authors.

They have addressed all my previous comments and concerns comprehensively and have made substantial improvements to the clarity, methodology, and overall presentation of the paper.

The authors’ responses were thorough and satisfactory, and the revisions have strengthened the manuscript significantly.

Therefore, I recommend accepting the manuscript for publication.

Best regards

Reviewer #2: The effort made in the revision is undeniable and respectable. Most of the points raised in the first round have been convincingly addressed.

However some have not. These are:

a) no reference or comment in the text on the diagram reported in figure 1 is reported despite the author's comment stating otherwise.

b) the empirical part is still poorly sound: the only innovation reported in this round is the performance of the VIF test for testing variables' collinearity which was not the point of the previous review. The methodology is a IV regression with count dependent variable. The reader needs to know that, for instance, a overdispersion test (to show the preferrability of negative binomial vs poisson) would be informative. No information on the goodness of fit of the two stages of the IV regression is provided. No test on the instruments adopted is available.

c) the information added in response to previous point 2 needs a revision. In the paragraph reported below, the first 2/3 periods need references:

"Few studies have examined the relationship between an integrated perspective and the moderating factors in the impact of OI on technological innovation performance, and the results are inconsistent. Firm must invest resources to absorb knowledge outflow. Among invested resources, internal R&D investment has been considered the most important resource for creating new knowledge and absorptive capacity. Internal R&D investment refers to the extent to which a firm invests in internal activities (e.g., new product development) and research and development resources, and internal R&D investment is said to be important in developing technological knowledge that contributes to absorptive capacity [19]."

7. PLOS authors have the option to publish the peer review history of their article (what does this mean?). If published, this will include your full peer review and any attached files.

Reviewer #1: No

Reviewer #2: No

---

## [Author Response · Author response to Decision Letter 1]

1 Aug 2024

Response to Reviewer #1 Comments

Point 1: I have carefully reviewed the revised manuscript submitted by the authors. They have addressed all my previous comments and concerns comprehensively and have made substantial improvements to the clarity, methodology, and overall presentation of the paper. The authors’ responses were thorough and satisfactory, and the revisions have strengthened the manuscript significantly. Therefore, I recommend accepting the manuscript for publication.

Response 1: Thank you for your recommendation for publication. We believe that your valuable comments have helped us to improve the manuscript.

Response to Reviewer #2 Comments

Thank you very much for all your comments. We are very glad that you give us the opportunity to revise our manuscript. We have worked hard to address your excellent and detailed comments. Based on them, we have substantially improved this manuscript. We hope you like the changes we made.

Point 1: no reference or comment in the text on the diagram reported in figure 1 is reported despite the author's comment stating otherwise.

Response 1: Thank you for your comment. As pointed out, this paper reflects the references or explanations in the text of the diagram reported in Figure 1.

Modification #1 (line 194, 236-237, 283, 332, 387, 420, line 424- 444 in revised manuscript)

The above figure structures the study's hypotheses in Figure 1. This study investigates the relationship between inbound open innovation (technology purchasing and technology collaboration) and innovation performance (patents) of biopharmaceutical firms, and what moderating factors are related between innovation performance (patents) and outbound open innovation (technology transfer). It is schematized. In addition, this study hypothesizes the moderating effects of government R&D support, absorptive capacity, and alliance management capacity on the relationship between inbound open innovation and innovation performance, and the relationship between innovation performance and outbound open innovation (technology transfer) as a moderating factor. set. Since innovation depends on a firm's ability to make external linkages and manage the innovation process [8], we propose that two specific types of organizational capabilities, namely alliance management capabilities and absorptive capabilities, will affect innovation performance. The government's R&D support serves as a source of funds for initial technology investment by firms with insufficient funds, promotes external cooperation or financing, and indirectly strengthens the company's R&D alliance by strengthening the company's absorptive capacity [77, 78]. On the other hand, companies that receive government R&D support have a crowding-out effect in which firms own R&D investment is replaced by government R&D support, showing a negative relationship [79]. Additionally, there is a possibility of moral hazard in using government R&D support for purposes other than research and development. There are also studies that show that there is no significant relationship between government R&D support and innovation performance. Therefore, efforts are needed to break away from the mixed results between government support and innovation performance and find better outcome variables that can verify the effectiveness of government R&D support. Additionally, strengthening a firm’s absorptive capacity increases the possibility of strategic alliances with various organizations [80]. Therefore, this study investigated how absorptive capacity, government R&D support, and alliance management capacity influence the moderating factors in the relationship between open innovation and performance, focusing on the Korean biopharmaceutical industry, a catch-up country.

Point 2: the empirical part is still poorly sound: the only innovation reported in this round is the performance of the VIF test for testing variables' collinearity which was not the point of the previous review. The methodology is a IV regression with count dependent variable. The reader needs to know that, for instance, a overdispersion test (to show the preferrability of negative binomial vs poisson) would be informative. No information on the goodness of fit of the two stages of the IV regression is provided. No test on the instruments adopted is available.

Response 2: Thanks for your sharp observation. We additionally reflected the content and results you mentioned in the manuscript. (line 562 – line 568 in revised manuscript)

In this study, the negative binomial regression results showed that the log likelihoods were -2063.3818, AIC (Akaike Information Criterion) was 4164.764, and BIC (Bayes Information Criterion) was 4261.851. The Poisson binomial regression results showed that the Log likelihoods were -2084.8472, AIC (Akaike Information Criterion) was 5783.694, and BIC (Bayes Information Criterion) was 5870.562. Therefore, in this study, an over-identified test was additionally performed after 2SLS. The F-test statistic value is 3.72096, which shows a weak correlation. As a result of the over-identification test, the p values of Sagan and Basmann (p = 0.0916, p = 0.0938) are greater than 0.05, so the instrumental variables have no correlation with the error term at the 5% significance level.

Point 3: the information added in response to previous point 2 needs a revision. In the paragraph reported below, the first 2/3 periods need references:

"Few studies have examined the relationship between an integrated perspective and the moderating factors in the impact of OI on technological innovation performance, and the results are inconsistent. Firm must invest resources to absorb knowledge outflow. Among invested resources, internal R&D investment has been considered the most important resource for creating new knowledge and absorptive capacity. Internal R&D investment refers to the extent to which a firm invests in internal activities (e.g., new product development) and research and development resources, and internal R&D investment is said to be important in developing technological knowledge that contributes to absorptive capacity [19]."

Response 3: Thanks for your sharp observation. We summarized and supplemented the various existing studies related to open innovation in biopharmaceutical firms. 

Modification #1 (line 106 – line 111 in revised manuscript)

Existing studies related to open innovation in the biopharmaceutical industry categorize open innovation types from a strategic perspective, open innovation incentives [20, 21], open innovation targets in the value chain, and The resulting relationship with technological innovation performance [22, 23, 24], the complementary resources or partnership experience of the two companies, differences in knowledge base, absorptive capacity, and government R&D support. The differences in performance and the choice of open innovation type were emphasized [19, 25, 26, 27, 28, 29].

In a study by Bianchi et al [20], depending on the stage of the development process, biopharmaceutical companies aim to acquire technology and knowledge through licensing agreements, alliances, etc. (inbound open innovation) or utilize it commercially (outbound open innovation). They said they are establishing increasingly intensive relationships with a variety of partners (e.g. large pharmaceutical companies, biotechnology companies, universities, etc.). A study by Allarakhia et al [21] found that open knowledge networks and other collaborative strategies give biopharmaceutical companies access to immaterial knowledge-based resources that are important for downstream drug development, and that these collaborative strategic alliances enable researchers to develop commercial products. When production is impossible and the costs associated with excessive upstream competition are too high, companies can jointly obtain incentives through collaborative knowledge production and open knowledge dissemination.

A study by Wang & Zajac [25] found that in the biopharmaceutical industry, the higher the similarity in resources and capabilities between two companies, the more likely it is to trigger companies to choose an acquisition as a governance form of resource combination rather than an alliance. A study by Shin et al [22] empirically analyzed the impact on technological innovation performance by type of alliance partner of biopharmaceutical companies and classified strategic alliances for R&D activities in the biopharmaceutical industry into three types to determine absorption capacity and potential competition. The moderating effect was identified. Vertical alliances have a positive effect on technological innovation performance, horizontal alliances have been shown to have an inverted U-shaped relationship with technological innovation performance due to the influence of competition, and the R&D intensity of biotechnology companies has a positive effect on technological innovation performance. -It was confirmed that there is a moderating effect that increases the impact of upstream alliances. A study by Baum et al [24] investigated the impact of changes in the alliance network composition of Canadian biotechnology startup companies on initial performance. They suggest that startups can improve their initial performance by building alliances, organizing them into efficient networks that provide access to diverse information and capabilities with minimal redundancy, conflict, and complexity costs, and by carefully forming alliances with potential competitors. . A study by Kang & Park [23] investigated the effect of cooperation between pharmaceutical and bio companies and the direct and indirect effects of government R&D support on innovation performance. According to the research results, upstream partnerships were significantly related to companies' innovation performance, and government R&D support directly and indirectly affected companies' innovation by promoting internal R&D and domestic upstream and downstream cooperation. The importance of government R&D support and networking and cooperation between universities, research institutes, and subparts was emphasized. A study by Carayannopoulos, & Auster [30] found that biopharmaceutical companies are more likely to source external knowledge through acquisition when the knowledge area is more complex and valuable when choosing acquisition or alliance when sourcing external knowledge, and that there is a higher possibility of sourcing external knowledge through alliance. The relationship between the two was also said to be strengthened.

Lin et al [26] investigated the impact on innovation performance from the perspective of inter-firm R&D alliance experience and absorptive capacity as an essential mechanism for creating new technological knowledge. Firms with high absorptive capacity and firms with more alliance experience show more innovative performance, and in particular, innovation performance peaks when the technological distance from the alliance partner is at a medium level when interacted with the proportion of R&D alliances in the firm's alliance portfolio. did. In addition, it was said that R&D alliances complement rather than replace internal R&D within a company. A study by Xia & Roper [27] investigated the impact of the relationship between absorptive capacity and external relationships, two key aspects of open innovation, on the growth of small biopharmaceutical companies in the United States and Europe. Research results show that absorptive capacity plays an important role in a company's growth, and that exploratory relationships are largely dependent on the continuity of R&D in terms of a company's absorptive capacity and interaction with the outside world. On the other hand, participation in exploitative relationships is related to the company's absorptive capacity. It was said that there were more conditions regarding competency.

According to George et al [28], pharmaceutical and bio companies' alliance portfolio characteristics and absorptive capacity together affect the company's innovative and financial performance. Lu et al [29] classified them into inbound OI and outbound OI, respectively. The impact on a company's innovation performance was studied. Research results show that inbound and outbound OI have a positive effect on a company's innovation performance, and absorptive capacity positively regulates inbound and outbound OI and a company's innovation performance. As such, a variety of existing studies have been conducted on open innovation in biopharmaceutical companies, but the studies that have been conducted so far are fragmented and do not present integrated results. In addition, existing studies were mainly conducted in advanced countries in the biopharmaceutical industry, necessitating caution in interpreting the implications of open innovation in catching-up countries. In this regard, this study investigated how absorptive capacity, government R&D support, and alliance management capacity affect the moderating factors in the relationship between open innovation and performance, focusing on the Korean pharmaceutical and bio industry, a catch-up country. We aim to derive managerial and policy implications for open innovation in the biopharmaceutical industry by applying it to Korean companies, which are catching up in the biopharmaceutical industry, and analyzing and comparing them.

---

## [Editor Report · Decision Letter 2]

7 Aug 2024

PONE-D-23-42382R2Does the impact of open innovation depend on contextual factors? A case of the Korean biopharmaceutical industryPLOS ONE

Dear Dr. Shin,

Thank you for submitting your manuscript to PLOS ONE. After careful consideration, we feel that it has merit but does not fully meet PLOS ONE’s publication criteria as it currently stands. Therefore, we invite you to submit a revised version of the manuscript that addresses the points raised during the review process.

Dear Authors,

we received the reports from the reviewers. One of the reviewer still requires some revisions. I kindly ask you to revise the manuscript accordingly.

Best,

EP

We look forward to receiving your revised manuscript.

Kind regards,

Eleonora Pierucci

Academic Editor

PLOS ONE

Additional Editor Comments:

Dear Authors,

we received the reports from the reviewers. One of the reviewer still requires some revisions. I kindly ask you to revise the manuscript accordingly.

Best,

EP

---

## [Author Response · Author response to Decision Letter 2]

19 Aug 2024

Response to Reviewer #2 Comments (3rd Round)

Thank you very much for all your comments. We are very glad that you give us the opportunity to revise our manuscript. We have worked hard to address your excellent and detailed comments, to supplement the shortcomings of our article. Based on them, we have substantially improved this manuscript. We hope you like the changes we made.

Point 1: no reference or comment in the text on the diagram reported in figure 1 is reported despite the author's comment stating otherwise.

Response 1: Thank you for your comment. As pointed out, this paper reflects the references or explanations in the text of the diagram reported in Figure 1. In this study, the research hypothesis is shown in Figure 1. In the figure, this study investigates the relationship between inbound open innovation (technology purchase and technology collaboration) and innovation performance (patents) of biopharmaceutical firms, and diagrams what moderating factors(Government R&D support, Absorptive capacity, Alliacne management capability) are related to innovation performance (patents) and outbound open innovation (technology transfer).

Modification #1 (line 235, 277-278, 324, 373, 428, 462, line 467- 544 in revised manuscript)

Figure 1. Schematic diagram of research hypothesis

The above figure structures the study's hypotheses in Figure 1. This study investigates the relationship between inbound open innovation (technology purchasing and technology collaboration) and innovation performance (patents) of biopharmaceutical firms, and what moderating factors are related between innovation performance (patents) and outbound open innovation (technology transfer). Technology purchasing is important for R&D-intensive (high-tech) firms because they have a high demand for innovation [36]. Firms can attract R&D investment through technology purchases and utilize other organizations’ resources and capabilities, including technology [38], It can develop internal innovation processes by integrating and utilizing external prepared technologies to address market gaps [42]. Technology purchasing can improve sales generation and financial performance by shortening the time for new product development and creating innovation outcomes [40], as it allows companies to develop complex products through the integration of proven technologies [41].

This requires cooperation with external partners [45], meaning that the role of OI is increasing for firms. Through OI activities, that is, collaboration, firms can provide access to scarce knowledge and technologies, reduce development costs, provide risk-sharing possibilities, and improve product-development processes [46]. Collaboration represents a distinct type of open innovation because it involves mutual innovation activities with common goals and the active participation of external stakeholders [71]. High-tech firms engage in extensive collaboration to secure external knowledge and accelerate technological innovation. Furthermore, as the capabilities and knowledge required for development are distributed across firms and institutions (universities), they collaborate with external partners. Collaboration increases the likelihood of goal achievement by securing additional resources and avoiding negative contingencies [48]. Such collaboration promotes innovation as a driving force for knowledge production and the creation of new innovations and expands a firm’s knowledge base that can be exploited for knowledge redistribution or transfer.

In particular, in the biopharmaceutical industry, open innovation is no longer an option but an essential strategic measure. Pisano [14] finds the reason in the characteristics of biopharmaceutical [14]. First, biopharmaceutical has high uncertainty, second, it has multidisciplinary characteristics, and third, it requires technological accumulation. Because of the high uncertainty of technology, biopharmaceutical firms proceed with one or two highly certain candidate substances in the pipeline, and also do not solve all processes from candidate substance discovery to clinical trial performance on their own, but rather choose to divide the work by stage. Furthermore, since biopharmaceutical firms often base their efforts on one or two accumulated core technologies, they must integrate complementary technologies from various fields through open innovation in order to develop them into commercial technologies [36]. In other words, because pharmaceutical and biotechnology requires accumulated capabilities through numerous failures, it is impossible to develop technologies from various fields together, which means that open innovation is inevitable. These industrial characteristics provide pharmaceutical and biotechnology firms with strategic advantages such as avoiding high costs for internal development, achieving rapid growth [42], and accessing cutting-edge technologies through technology purchases and collaborations with various research institutes such as universities, hospitals, and government-funded research institutes as well as pharmaceutical firms [19, 23]. Therefore, biopharmaceutical firms can select excellent technologies from biotechnology firms in advance, reduce the risk of failure during development, and thereby increase the R&D efficiency of biotechnology firms, thereby promoting technological innovation results.

In addition, through this opportunity to acquire and learn complementary resources and capabilities from external organizations, biopharmaceutical firms can receive positive impacts, such as R&D performance and patent performance [24]. In addition, excellent scientific knowledge or basic technology from research institutes and biotechnology firms is transferred to pharmaceutical firms, which helps in terms of commercialization, improved financial performance, and technological innovation performance. It is schematized. In addition, this study hypothesizes the moderating effects of government R&D support, absorptive capacity, and alliance management capacity on the relationship between inbound open innovation and innovation performance, and the relationship between innovation performance and outbound open innovation (technology transfer) as a moderating factor. set. Since innovation depends on a firm's ability to make external linkages and manage the innovation process [8], we propose that two specific types of organizational capabilities, namely alliance management capabilities and absorptive capabilities, will affect innovation performance. The government's R&D support serves as a source of funds for initial technology investment by firms with insufficient funds, promotes external cooperation or financing, and indirectly strengthens the company's R&D alliance by strengthening the company's absorptive capacity [77, 78]. On the other hand, companies that receive government R&D support have a crowding-out effect in which firms own R&D investment is replaced by government R&D support, showing a negative relationship [79]. Additionally, there is a possibility of moral hazard in using government R&D support for purposes other than research and development. There are also studies that show that there is no significant relationship between government R&D support and innovation performance. Therefore, efforts are needed to break away from the mixed results between government support and innovation performance and find better outcome variables that can verify the effectiveness of government R&D support. Additionally, strengthening a firm’s absorptive capacity increases the possibility of strategic alliances with various organizations [80]. Therefore, this study investigated how absorptive capacity, government R&D support, and alliance management capacity influence the moderating factors in the relationship between open innovation and performance, focusing on the Korean biopharmaceutical industry, a catch-up country.

Reference

8. Chesbrough, H., & Bogers, M. Explicating open innovation: Clarifying an emerging paradigm for understanding innovation. New Frontiers in Open Innovation. Oxford: Oxford University Press, Forthcoming, 2014; 3-28.

14. Lyu, J., Wang, S., Balius, T. E., Singh, I., Levit, A., Moroz, Y. S., ... & Irwin, J. J. Ultra-large library docking for discovering new chemotypes. Nature.2019;566:, 224-229.

19. Cohen WM, Levinthal DA. Absorptive capacity: A new perspective on learning and innovation. Admin Sci Q. 1990;35: 128-152. doi: 10.2307/2393553

23. Kang KN, Park H. Influence of government R&D support and inter-firm collaborations on innovation in Korean biotechnology SMEs. Technovation. 2012;32: 68-78. doi: 10.1016/j.technovation.2011.08.004.

24. Baum JAC, Calabrese T, Silverman BS. Don't go it alone: Alliance network composition and startups' performance in Canadian biotechnology. Strateg Manag J. 2000;21: 267-294. doi: 10.1002/(SICI)1097-0266(200003) 21:3<267::AID-SMJ89>3.0.CO;2-8.

36. Rothaermel FT, Deeds DL. Alliance type, alliance experience and alliance management capability in high-technology ventures. J Bus Venturing. 2006;21: 429-460. doi: 10.1016/j.jbusvent.2005.02.006.

38. Rothaermel FT. Complementary assets, strategic alliances, and the incumbent’s advantage: An empirical study of industry and firm effects in the biopharmaceutical industry. Res Policy. 2001;30: 1235-1251. doi: 10.1016/S0048-7333(00)00142-6.

40. Wang Y, Vanhaverbeke W, Roijakkers N. Exploring the impact of open innovation on national systems of innovation — A theoretical analysis. Technol Forecasting Soc Change. 2012;79: 419-428. doi: 10.1016/j.techfore.2011.08.009.

41. Van de Vrande V, De Jong JPJ, Vanhaverbeke W, De Rochemont M. Open innovation in SMEs: Trends, motives and management challenges. Technovation. 2009;29: 423-437. doi: 10.1016/j.technovation.2008.10.001.

42. Chesbrough H, Vanhaverbeke W, West J, editors. Open innovation: Researching a new paradigm. Oxford University Press; 2006.

45. Frishammar J, Parida V, Westerberg M. Inbound open innovation activities in high‐tech SMEs: The impact on innovation performance. J Small Bus Manag. 2012;50: 283-309. doi: 10.1111/j.1540-627X.2012.00354.x. 

46. Henkel J. Selective revealing in open innovation processes: The case of embedded Linux. Res Policy. 2006;35: 953-969. doi: 10.1016/j.respol.2006.04.010

48. Audretsch DB, Belitski M. The role of R&D and knowledge spillovers in innovation and productivity. Eur Econ Rev. 2020;123: 103391. doi: 10.1016/j.euroecorev.2020.103391.

71. Enkel E, Gassmann O, Chesbrough H. Open R&D and open innovation: Exploring the phenomenon. R D Manag. 2009;39: 311-316. doi: 10.1111/j.1467-9310.2009.00570.x.

77. Sakakibara, M. Heterogeneity of firm capabilities and cooperative research and development: an empirical examination of motives. Strategic management journal. 1997;18:143-164. doi: 10.1002/(sici)1097-0266(199707)18:1+ <143::aid-smj927>3.0.co;2-y

78. Svensson, R. Innovation performance and government financing. Journal of small business & Entrepreneurship.2008; 21: 95-116. doi:10.1080/08276331.2008.10593415

79. Luukkonen, T.The difficulties in assessing the impact of EU framework programmes. Research Policy. 1998; 27: 599-610. doi:10.1016/s0048-7333(98)00058-4

80. Muscio, A. The impact of absorptive capacity on SMEs' collaboration. Economics of Innovation and New Technology. 2007;16: 653-668. doi:10.1080/10438590600983994

Point 2: 

Point 2a: the empirical part is still poorly sound: the only innovation reported in this round is the performance of the VIF test for testing variables' collinearity which was not the point of the previous review. The methodology is a IV regression with count dependent variable. The reader needs to know that, for instance, a overdispersion test (to show the preferrability of negative binomial vs poisson) would be informative. 

Response 2a: Thanks for your comment. In our study, we removed the performance results of the VIF test for testing the collinearity of variables. In addition we conducted an overdispersion test (to show the preference of negative binomial vs. Poisson) in the methodology. The results reflect the basis and conclusion that the negative binomial regression is an appropriate analysis model. We additionally reflected the content and results you mentioned in the manuscript. (line 663 – line 693 in revised manuscript)

In this study, negative binomial regression and Poisson binomial regression tests were performed, and it was determined that negative binomial regression analysis was an appropriate analysis model because the log likelihood value (-2063.3818), AIC value (4164.764), and BIC value (4261.851) were smaller than those of the Poisson analysis results. The detailed methods and results are as follows.

The Poisson regression rarely fits in practice since in most applications the conditional variance is greater than the conditional mean. If the mean structure is correct, but inefficient (Gourierourx et al., 1984). Further, the standard errors from the Poisson regression model will be biased downward, resulting in spuriously large z-values (Cameron and Trivedi 1986, p. 31).

In this study, Negative binomial regression and Poisson binomial regression tests were performed to obtain information on the goodness of fit of the two stages of the IV regression. The negative binomial regression results showed that the Log likelihood value was –2063.3818, the AIC (Akaike Information Criterion) was 4164.764, and the BIC (Bayes Information Criterion) was 4261.851. The Poisson binomial regression results showed that the Log likelihood value was –2084.8472, the AIC (Akaike Information Criterion) was 5783.694, and the BIC (Bayes Information Criterion) was 5870.562. 

Also, the likelihood-ratio test is a test for the overdispersion parameter alpha (our results show that the Likelihood-ratio test of alpha=0: chibar2(01) = 1558.42 Prob>=chibar2 = 0.000). When the overdispersion parameter is 0, the negative binomial distribution is identical to the Poisson distribution. In this case, alpha is significantly different from 0, so we emphasize once again that the Poisson distribution is not appropriate. In other words, Testing for Overdispersion (a test to statistically justify why Negative binomial regression vs Poisson binomial regression model is used) rejected Ho within a statistically significant range, so NBR was used rather than the Poisson regression model.

AIC (Akaike Information Criterion) and BIC (Bayes Information Criterion) are standard measures for model selection, and a regression equation with a smaller value, whether it is the AIC value or the BIC value, is a more appropriate regression equation. In other words, a smaller AIC and BIC means that the model has the largest degree of coupling (likelihood) and the smallest number of variables. In this study, considering the analysis results, negative binomial regression was judged to be an appropriate analysis model.

Point 2b: No information on the goodness of fit of the two stages of the IV regression is provided. No test on the instruments adopted is available.

Response 2b: Thanks for your sharp observation. We additionally reflected the content and results you mentioned in the manuscript. (line 694 – line 706 in revised manuscript)

In addition, after performing 2SLS, an estimation method using instrumental variables (IV) was implemented, and the F test statistic was less than 10 (F test statistic = 3.72096), so it was judged that the correlation with the endogenous variable was weak. The detailed explanation and method are as follows.

In this study, an estimation method using instrumental variables (IV) was implemented after performing 2SLS. Instrumental variables must not necessarily be correlated with the error term of the regression model and must be correlated with the endogenous explanatory variables. In IV estimation, the number of instrumental variables must be greater than or equal to the number of endogenous explanatory variables, and when there are more instrumental variables than endogenous explanatory variables, it is called over-identification. In the case of over-identification, it is necessary to test the validity of the instrumental variables, that is, to conduct an over-identification test. Therefore, in this study, an over-identificat

---

## [Editor Report · Decision Letter 3]

26 Aug 2024

Does the impact of open innovation depend on contextual factors? A case of the Korean biopharmaceutical industry

PONE-D-23-42382R3

Dear Dr. Shin,

We’re pleased to inform you that your manuscript has been judged scientifically suitable for publication and will be formally accepted for publication once it meets all outstanding technical requirements.

Kind regards,

Eleonora Pierucci

Academic Editor

PLOS ONE
---

## [Editor Report · Acceptance letter]

30 Aug 2024

PONE-D-23-42382R3 

PLOS ONE

Dear Dr. Shin, 

I'm pleased to inform you that your manuscript has been deemed suitable for publication in PLOS ONE. Congratulations! Your manuscript is now being handed over to our production team.

Kind regards, 

on behalf of

Professor Eleonora Pierucci 

Academic Editor

PLOS ONE